# What do Graph Neural Networks learn? Insights from Tropical Geometry

**Tuan Anh Pham**
School of Mathematics
University of Edinburgh
Edinburgh, United Kingdom
`tuan.pham@ed.ac.uk`

**Vikas Garg**
YaiYai Ltd and Aalto University
`vgarg@csail.mit.edu`

## Abstract

Graph neural networks (GNNs) have been analyzed from multiple perspectives, including the WL-hierarchy, which exposes limits on their expressivity to distinguish graphs. However, characterizing the class of functions that they learn has remained unresolved. We address this fundamental question for message passing GNNs under ReLU activations, i.e., the de-facto choice for most GNNs.

We first show that such GNNs learn tropical rational signomial maps or continuous piecewise linear functions, establishing an equivalence with feedforward networks (FNNs). We then elucidate the role of the choice of aggregation and update functions, and derive the first general upper and lower bounds on the geometric complexity (i.e., the number of linear regions), establishing new results for popular architectures such as GraphSAGE and GIN. We also introduce and theoretically analyze several new architectures to illuminate the relative merits of the feedforward and the message passing layers, and the tradeoffs involving depth and number of trainable parameters. Finally, we also characterize the decision boundary for node and graph classification tasks.

## 1 Introduction

Message passing has been a cornerstone of machine learning, from inference in graphical models [1, 2] to embedding of graphs [3, 4, 5, 6, 7]. Message passing neural networks (MPNNs) are easy to implement, and can handle large scale, heterogeneous, and dynamic real world data effectively; so continue to be an active area of research [8, 9, 10]. Indeed, several popular GNNs are usually cast and implemented as MPNNs, see e.g., [11, 12, 13, 14, 15].

Unsurprisingly, GNNs as MPNNs have been analyzed from multiple theoretical perspectives. A major theme is inspired by connections to the so-called 1-WL (Weisfeiler-Leman) test for group isomorphism and its higher order extensions, where nodes in a (hyper-)graph repeatedly refine their *colors* based on the messages from their neighbors [16]. MPNNs are known to be bounded in power according to the WL-hierarchy [15, 17, 18], implying their inability to distinguish some non-isomorphic graphs. Unlike WL that strives to expose what GNNs *cannot* do, here we seek to unravel what they *can*.

Notably, the WL formalism implicitly relies on injective hash functions; in contrast, most successful GNNs typically use ReLU activations that violate injectivity. Bounding the expressivity of such GNNs via their injective surrogates is theoretically valid; however, it does not illuminate what they learn. Indeed, several fundamental questions remain elusive for these practical models; e.g., (a) what class of functions can they represent, (b) how does their expressivity vary with the choice of message aggregation and update functions, (c) what complexity tradeoffs (e.g., in terms of the number of

38th Conference on Neural Information Processing Systems (NeurIPS 2024).

| Theoretical contributions of this work | |
|---|---|
| **Characterizing the class of functions learned by ReLU MPNNs:** | |
| Equivalence with ReLU FNNs, TRSMs and CPLMs | Proposition 1 |
| **Estimating the number of linear regions, and complexity tradeoffs:** | |
| First general lower bound for ReLU MPNNs | Theorem 3 |
| First general upper bound for ReLU MPNNs | Theorem 4 |
| Max aggregation has greater geometric complexity than sum | Proposition 5 |
| Recovery of existing upper bounds for FNN and GCN | Corollary 1, 2 |
| New upper bounds for GraphSAGE and GIN | Corollary 3, 4 |
| **New ReLU MPNNs and complexity tradeoffs:** | |
| New architectures that can all learn CPLMs, and their tradeoffs | Proposition 6 |
| **Characterizing the decision boundary:** | |
| Decision boundary of ReLU MPNNs for graph classification | Proposition 7 |
| Decision boundary of ReLU MPNNs for node classification | Proposition 8 |

Figure 1: **Overview of our results**. Formulating ReLU MPNNs as tropical geometric objects allows us to shed light on several important aspects where WL falls short.

layers or parameters) exist for models of comparable expressivity, and (d) what decision boundary emerges for node and graph classification tasks?

We appeal to tropical algebra and geometry to address these questions in the context of ReLU MPNNs. Specifically, casting these networks as tropical geometric objects, we analyze their ability to represent different weighted sums of *tropical monomials* (the basic objects of interest in tropical algebra, akin to monomials for standard polynomials) at different nodes. We first show that the family of functions represented by these networks is precisely the family of *tropical rational signomial maps* (TRSMs). Since TRSMs are known to be equivalent to ReLU FNNs [19], this reveals an equivalence between ReLU MPNNs and ReLU FNNs: they represent exactly the same class of functions, namely, continuous piecewise linear maps (CPLMs).

This equivalence in terms of expressivity does not, however, translate into parity in *efficiency*, e.g., as quantified in terms of their respective requirements for the number of layers and trainable parameters to represent an arbitrary TRSM. Indeed, MPNNs have some strengths and limitations relative to FNNs. On the positive side, MPNNs benefit from parallel computation and streamlined communication between the nodes and their neighbors. On the flip side, MPNNs are constrained by permutation-equivariant layers that employ permutation-invariant aggregation operators [20], which impedes their ability to represent arbitrary combinations of tropical monomials.

In order to better understand the efficiency of different MPNN architectures, we investigate their *geometric complexity*, the number of linear regions that they can distinguish. Towards that end, we establish the first general upper and lower bounds on the geometric complexity of ReLU MPNNs, recovering the existing results [19, 21] for FNNs and graph convolutional networks (GCNs) as special cases. Importantly, we also unravel new results for two of the most prominent MPNNs whose complexity was previously unknown, namely, GraphSAGE [13] and GIN [15].

A particularly attractive aspect of our bounds is manifested in segregation of the contribution from different components, such as aggregation and update steps. In particular, they reveal that selecting *coordinate-wise max* as the aggregation operator affords greater geometric complexity than *sum* for ReLU MPNNs. In order to provide further insights about various complexity tradeoffs, we introduce and analyze four novel MPNN abstractions. Notably, they can all represent arbitrary TRSMs, i.e., they are as expressive as ReLU FNNs, but differ in terms of layers as well as total learnable parameters in their corresponding architectures. Fundamentally, we expose a general trend about the relative merits of the feedforward and message passing paradigms: fewer layers for MPNNs (particularly with coordinate-wise max aggregation) but fewer trainable parameters for FNNs.

Finally, we study the decision boundaries for node and graph classification, uncovering the underlying connections with *tropical hypersurfaces*; i.e., the set of points where two or more tropical monomials achieve the same value.

We summarize our key contributions in Figure 1. We now proceed to reviewing some related works.

## 1.1 Related work

**Tropical Geometry and Machine Learning.** Tropical geometry [22, 23] provides tools to study the algebraic geometry and combinatorics of continuous piecewise linear functions, and finds several applications (e.g., in optimization). Two seminal works [19, 24] initiated the analysis of deep learning models via tropical geometry, establishing the link between ReLU FNNs and tropical rational functions. The connection was then extended to maxout-layers in [25]. The decision boundaries of FNNs through a tropical lens were studied in [26]. Other aspects of deep neural networks have also be analyzed [27, 28, 29, 30, 31, 32]. We refer to [33] for a survey on the current use of tropical geometry in deep learning.

**Expressivity and WL.** Much work on GNNs has been inspired by noticing the parallels between MPNNs and the WL test for isomorphism [15, 34, 35]. Standard MPNNs are no more powerful than 1-WL (equivalently, 2WL), so higher order models that consider tuples of vertices have been proposed [17, 18]. Several adaptations of WL such as geometric WL [36], cellular WL [37],temporal WL [10], and persistent WL [38] have been introduced in different contexts. Limited expressivity and symmetry considerations have led to the design of new MPNN architectures [15, 39, 40, 41, 42, 43], inclusion of different types of features [39, 44], and integration with topological descriptors [45, 46, 47]. Other notions of expressivity have also been proposed, e.g., equivariant polynomials [48] and homomorphism counts [49].

**Other results about GNNs.** GNNs as MPNNs are also known to be prone to information bottle-necks [50, 51, 52], oversmoothing and oversquashing [53], and heterophily [54, 55]. Some works have also established their inability to count substructures, compute graph properties, or learn topology [49, 56, 57]; connections to algebraic topology [58], biconnectivity [59], and communication complexity [60, 61], behavior in overparametrized regimes [62, 63]; power of recursion in counting substructures [64]; benefits of positional encoding [65]; and ability to generalize and achieve universal approximation [56, 66, 67, 68, 69, 70, 71, 72, 73, 74, 75].

However, neither expressivity in terms of WL nor these other results characterize the class of functions learned by GNNs, and in this work we bridge this glaring gap for ReLU MPNNs.

**Complexity of Neural Networks.** The complexity of deep neural networks is typically studied in terms of the number of linear regions [76, 77, 78, 79], and the intricacy of decision boundary [19], though other notions such as trajectory length [80] have also been considered recently. The number of linear regions quantifies the flexibility of the function class, thus bounding the number of regions relates closely to both expressivity and generalization [79]. To our knowledge, only the geometric complexity upper bounds for GCNs are known in the context of GNNs [21]. We provide the first general lower and upper bounds for ReLU MPNNs, establishing complexity of popular architectures such as GraphSage and GIN, and recovering the results for GCNs and FNNs as special cases.

## 2 Preliminaries

### 2.1 Message Passing Neural Networks (MPNNs)

A general $T$-layer MPNN $\chi$ is defined as a sequence of layers $\{\varphi^{(t)}\}_{t=1}^{T}$, and takes as input a (directed) graph $G = (V, E, X)$ with vertices $v \in V$, edges $(u, v) \in E \subseteq V \times V$, and features $X = (X_V, X_E)$ consisting of node attributes $X_V \in \mathbb{R}^{|V| \times d_0}$ and edge attributes $X_E \in \mathbb{R}^{|E| \times d_0'}$. It produces as output a refined *embedding* for each vertex and edge. Formally, $\chi$ acts on $X$ as

$$\chi(X) = \varphi^{(T)} \circ ... \circ \varphi^{(1)}(X) \,, \text{ where each } \varphi^{(t)} : \mathbb{R}^{|V| \times d_{t-1}} \times \mathbb{R}^{|E| \times d_{t-1}'} \to \mathbb{R}^{|V| \times d_t} \times \mathbb{R}^{|E| \times d_t'}$$

is *permutation-equivariant* [20]. Subsequently, depending on the task, the embedding for each node produced by $\chi$ is fed into another neural network, e.g., $\eta : \mathbb{R}^{d_T} \to \mathbb{R}^{out}$ for node classification or

regression (and similarly for edges). An additional readout step $\varphi_{\text{Readout}} : \mathbb{R}^{|V| \times d_T} \times \mathbb{R}^{|E| \times d'_T} \to \mathbb{R}^{d_G}$ is usually required to obtain a single vector from the output of $\chi$ for graph-level prediction.

We thus proceed to describing the working of each layer $\varphi^{(t)}$ that yields a representation $h_v^{(t)} \in \mathbb{R}^{d_t}$ for each vertex $v \in V$ and a representation $e_{uv}^{(t)} \in \mathbb{R}^{d'_t}$ for each edge $(u, v) \in E$. Assuming node embeddings are updated before edge embeddings (the converse works analogously), we have

$$h_v^{(t)} = \varphi_{\text{Update}}^{(t)}(h_v^{(t-1)}, m_v^{(t)}), \quad \text{where}$$

$$m_v^{(t)} = \varphi_{\text{Agg}}^{(t)}(h_v^{(t-1)}, \{\{h_u^{(t-1)}, e_{uv}^{(t-1)}, e_{vu}^{(t-1)} | u \in \text{Ne}(v)\}\}) .$$

*Aggregation functions* $\{\varphi_{Agg}^{(t)}\}$ are typically *permutation-invariant*, and we use an operator $\square$ such as sum, average, or coordinate-wise max/min to combine the *messages* [15, 81]:

$$\varphi_{Agg}^{(t)}(h_v^{(t-1)}, \{\{h_u^{(t-1)}, e_{uv}^{(t-1)}, e_{vu}^{(t-1)}\}\}_{u \in \text{Ne}(v)}) = \phi_1^{(t)}(\square_{u \in \text{Ne}(v)}^{(t)} \phi_2^{(t)}(h_v^{(t-1)}, h_u^{(t-1)}, e_{uv}^{(t-1)}, e_{vu}^{(t-1)})), \tag{1}$$

where $\{\phi_1^{(t)}\}$ and $\{\phi_2^{(t)}\}$ are usually implemented as FNNs, $\text{Ne}(v)$ denotes the set of neighboring vertices of $v$, and $\{\{\cdot\}\}$ denotes a multiset. When there is no confusion, we usually write $\square^{(t)}$ instead. Let $L_1^{(t)}$ and $L_2^{(t)}$ denote, respectively, the number of layers in $\phi_1^{(t)}$ and $\phi_2^{(t)}$. We use $n_1^{t,\ell}$ (and $n_2^{t,\ell}$) to denote the dimension of layer $\ell$ in $\phi_1^{(t)}$ (and $\phi_2^{(t)}$). On the other hand, the *update functions* take the form

$$\varphi_{\text{Update}}^{(t)}(h_v^{(t-1)}, m_v^{(t)}) = \sigma_{\text{Update}}^{(t)}(W_{\text{self}}^{(t)} h_v^{(t-1)} + W_{\text{neigh}}^{(t)} m_v^{(t)}).$$

Henceforth, we focus on sum and coordinate-wise max, since coordinate-wise min and average can be obtained from coordinate-max and sum respectively. We shall also number the vertices in $V$ with $A_1, ..., A_{|V|}$ and edges in $E$ with $e_1, ..., e_{|E|}$, and use the lexical order for edges, i.e., $e_{uv} < e_{u'v'} \Leftrightarrow u < u'$ or $(u = u'$ and $v < v') .$

## 2.2 Tropical Algebra

Here we adopt the notation from [19]. Let $\mathbb{T} = \mathbb{R} \cup \{-\infty\}$ be an extended set of real numbers. We equip $\mathbb{T}$ with two binary operators **tropical sum** $\oplus$ and **tropical multiplication** $\odot$:

$$a \oplus b = \max\{a, b\} ; \quad a \odot b = a + b ;$$

$$a \oplus -\infty = -\infty \oplus a = a ; \quad a \odot -\infty = -\infty \odot a = -\infty ,$$

where $\max$ and $+$ are the usual operators in $\mathbb{R}$. Thus, $(\mathbb{T}, \oplus, \odot)$ is a semi-ring with additive identity $-\infty$ and multiplicative identity $0$. We also define the **tropical power** $x^{\odot a}$ for each $x \in \mathbb{R}$:

$$x^{\odot a} = \begin{cases} x \odot ... \odot x = a \cdot x & \text{if } a \in \mathbb{N} \\ (-x)^{\odot(-a)} & \text{if } a \in \mathbb{Z} \setminus \mathbb{N} \end{cases} , \quad -\infty^{\odot a} = \begin{cases} \infty & \text{if } a \in \mathbb{N} \setminus \{0\} \\ 0 & \text{if } a = 0 \end{cases} ;$$

where $\cdot$ is the standard product over $\mathbb{R}$, and often abbreviate $x^{\odot a}$ to $x^a$ without inducing any confusion. A **tropical monomial** in $m$ variables takes the form $c \odot x_1^{a_1} \odot ... \odot x_m^{a_m}$ for $c \in \mathbb{T}$ and $a_1, ..., a_m \in \mathbb{N}$, and is often denoted by multi-index shorthand $cx^\alpha$, where $\alpha = (a_1, ..., a_m) \in \mathbb{Z}^m$ and $x = (x_1, ..., x_m) \in \mathbb{R}^m$. Note that this may be interpreted as the affine combination $x^\top \alpha + c$.

A **tropical polynomial** $f(x) = c_1 x^{\alpha_1} \oplus ... \oplus c_r x^{\alpha_r}$ is a finite tropical sum of tropical monomials, and amounts to a max over finitely many terms, i.e., $f(x) = \max_{i=1}^r \{x^\top \alpha_i + c_i\}$. Without loss of generality, we assume $\alpha_i \neq \alpha_j$ for $i \neq j$ since we can combine monomials with the same $\alpha$ into one. A **tropical rational function** $f \oslash g(x) = f(x) - g(x)$ is the difference between two tropical polynomials. A map $F : \mathbb{R}^m \to \mathbb{R}^p$ with each component a tropical polynomial [resp. tropical rational function] is called a **tropical polynomial map** [resp. **tropical rational map**], and belongs to the set $Pol(m, p)$ [resp. $Rat(m, p)$].

When we allow $\alpha_i \in \mathbb{R}^m$ instead of restricting its components to integers ($\alpha_i \in \mathbb{Z}^m$), we obtain a **tropical signomial function/map** [resp. **tropical rational signomial function/map**] instead of a tropical polynomial/map [resp. tropical rational function/map]. It is known that each tropical rational signomial map (TRSM) is a continuous piecewise linear map (CPLM) and vice versa [19].

## 3 Tropical Algebra of MPNNs

In this section, we characterize the class of functions learned by ReLU MPNNs, whose activation functions for both nodes and edges (i.e., those used for $\eta$, $\phi_1^{(t)}$, $\phi_2^{(t)}$, $\sigma_{\text{Update}}^{(t)}$, etc.) are of the form:

$$\sigma^{(l)}(x) = \max\{x, t^{(l)}\}, \text{ where } t^{(l)} \in (\mathbb{R} \cup -\infty)^{n_l}.$$

In particular, note that $\sigma^{(l)}(x) = x$ when $t^{(l)} = -\infty$. In contrast, $t^{(l)} = 0$ purges all negative inputs.

Let $\mathcal{F}_{\text{ReLU MPNN}}, \mathcal{F}_{\text{ReLU FNN}}, \mathcal{F}_{\text{CPLM}}, \mathcal{F}_{\text{TRSM}}$ be the set of functions represented by all ReLU MPNNs, ReLU FNNs, CPLMs and TRSMs respectively. [19] established the equivalence of ReLU FNNs, CPLMs and TRSMs.

**Lemma 1** ([19]). $\mathcal{F}_{\text{ReLU FNN}} = \mathcal{F}_{\text{CPLM}} = \mathcal{F}_{\text{TRSM}}$.

We will now extend this result to establish equivalence with $\mathcal{F}_{\text{ReLU MPNN}}$. Equating ReLU MPNNs with CPLMs is rather nuanced, since nodes in each layer share the weights. Therefore, we employ two reductions showing (1) every ReLU FNN can be cast as a ReLU MPNN, and (2) every ReLU MPNN, in turn, can be expressed as a TRSM.

**Proposition 1.** *[Equivalence of ReLU MPNNs, ReLU FNNs, TRSMs and CPLMs] $\mathcal{F}_{\text{ReLU MPNN}} = \mathcal{F}_{\text{ReLU FNN}} = \mathcal{F}_{\text{CPLM}} = \mathcal{F}_{\text{TRSM}}$. In other words, the following families are equivalent (with $m = |V|d + |E|d'$ and $p = |V|d_{out} + |E|d'_{out}$).*

1. *ReLU MPNNs $\chi : \mathbb{R}^{|V| \times d} \times \mathbb{R}^{|E| \times d'} \to \mathbb{R}^{|V| \times d_{out}} \times \mathbb{R}^{|E| \times d'_{out}}$;*
2. *Tropical rational signomial maps (TRSMs) $F \oslash G : \mathbb{R}^m \to \mathbb{R}^p$;*
3. *Continuous piecewise linear maps (CPLMs) $\psi : \mathbb{R}^m \to \mathbb{R}^p$;*
4. *ReLU FNNs $\nu : \mathbb{R}^m \to \mathbb{R}^p$.*

**Remark 1.** *In contrast to the WL test, which exposes the limitations of MPNNs via injective hash functions, Proposition 1 characterizes the class of functions that can be represented by MPNNs with ReLU activations (that are non-injective). Note, however, that Proposition 1 does not quantify how effective ReLU MPNNs are in representing CPLMs. Moreover, the equivalence between ReLU MPNNs and ReLU FNNs (according to Proposition 1) does not explain the observed empirical discrepancy between ReLU MPNNs and ReLU FNNs in practice. This motivates our subsequent analysis and results (Section 5 and Table 1), which investigate the benefits of ReLU MPNNs in terms of both the number of learnable parameters and the number of layers required to represent the same CPLM.*

## 4 Geometric Complexity of ReLU MPNNs

We now invoke tools from tropical algebraic geometry to study the geometric complexity of ReLU MPNNs, and provide some insights into the model architecture. Following [19], for a CPLM $f : \mathbb{R}^m \to \mathbb{R}^p$, we define its *linear degree* $\mathcal{N}(f)$ to be $K$, where $K$ is the least number of connected regions $\Omega_k$ of $\mathbb{R}^m$ such that the restriction $f|_{\Omega_k}$ is affine. Equivalently, following [82], we can also define $K = \mathcal{N}(f)$ as follows: $f : \mathbb{R}^m \to \mathbb{R}^p$ is a CPLM if $f$ is continuous and there exists a set $\{f_k : k \in \{1, ..., K\}\}$ of affine functions and maximal connected subsets $(\Omega_k)_{k=1}^K$ satisfying the following conditions:

$$\Omega_i \cap \Omega_j = \emptyset; \quad \bigsqcup_{k=1}^K \Omega_k = \mathbb{R}^m; \quad f|_{\Omega_k} = f_k.$$

Similarly, we define the *convex degree* $\mathcal{N}_c(f)$ where we require additionally that $\Omega_k$ is convex. We further define $\mathcal{N}_c(f|m')$ as the maximum convex degree across restrictions of $f$ to different $m'$-dimensional affine subspaces of $\mathbb{R}^m$, $m' \leq m$. We will analyze general upper and lower bounds on $\mathcal{N}(\chi)$ for a ReLU MPNN $\chi$. Recall our setting of MPNN layers from Section 2.1 consisting of $\phi_{\text{Agg}}^{(t)}$ and $\phi_{\text{Update}}^{(t)}$. Note that $n_1^{t,l}$ and $n_2^{t,l}$ denote the intermediate dimensions in $\phi_1^{(t)}$ and $\phi_2^{(t)}$. Let $\tilde{d}_t$ and $\bar{d}_t$ be the output dimension of $\phi_2^{(t)}$ and $\phi_1^{(t)}$, thus $n_2^{t,L_2^{(t)}} = \tilde{d}_t$.

To analyze the number of linear region for a ReLU MPNN $\chi$, we make a simplifying assumption that all input graphs to $\chi$ have the same graph structure $G$. In particular, we denote the sum of degrees of all vertices in $G$ by $D$, and the maximum degree by $S$. A key step in our analysis of geometric

complexity is building a ReLU FNN that can be applied on the vectorized input $H^{(t-1)}$ of all node embeddings $h_v^{(t-1)}$. Using the notation indicated in Equation 1, we form a ReLU FNN $\Phi_2^{(t)}$ that achieves the same effect as $\phi_2^{(t)}$ for every adjacent node embedding. The aggregation operator can be seen as either a matrix multiplication for sum, or a FNN $\Phi_3^{(t)}$ for max aggregation - the difference between these two cases will be discussed in Section 4.3. Similarly, we form a ReLU FNN $\Phi_1^{(t)}$ for $\phi_1^{(t)}$, which in combination with $\Phi_2^{(t)}$ and $\Phi_3^{(t)}$ can be seen as a ReLU FNN $\Phi_{\text{Agg}}^{(t)}$.

Our next proposition is important in the analysis for geometric complexity of $\chi$, as it relates the geometric complexity of the model up to the $t+1$-th layer to that of the $t$−th layer, and the update as well as the message components. We define the vectorized concatenated embedding of all the vertices in the $t$-th layer to be $H^{(t)} \in \mathbb{R}^{|V|d_t}$ and show that the result of $t$-th message aggregation step can be written as a result of a FNN $\Phi_{\text{Agg}}^t(H^{(t)})$. Modifying any aggregation or update component thus will affect the bound and in particular, the choice of aggregation operator has an impact on the geometric complexity (which we will discuss in Section 4.3).

**Proposition 2.** *[Recursive formula for geometric complexity]*

$$\mathcal{N}(\varphi^{(t+1)}) \leq \mathcal{N}_c(\varphi^{(t+1)}) \leq \mathcal{N}_c(\varphi_{Update}^{(t+1)}||V|(d_t + \tilde{d}_{t+1}))\,\mathcal{N}_c(\Phi_{Agg}^{(t+1)}||V|d_t)\,\mathcal{N}_c(\varphi^{(t)}). \quad (2)$$

The ideas and proofs for Proposition 2 build on [19], and the details can be found in the Appendix.

## 4.1 Lower bound of geometric complexity

We note that the lower bound on the linear degree of $\chi$ depends heavily on the choice of weights and biases; e.g., setting their value to 0 trivially results in $\mathcal{N}(\chi) = 0$. Thus, a lower bound on the geometric complexity (i.e., maximal linear degree) is of greater interest. In this subsection, we will provide a general lower bound for the maximum number of linear regions for a ReLU MPNN, building on the work of [76].

**Theorem 3.** *[Lower bound on the maximum number of linear regions] Assume for all $t, l$, we have $n_1^{t,l}, n_2^{t,l} \geq d_0$ and let $n_{1,d_0}^{t,l} = \left\lfloor \frac{n_1^{t,l}}{d_0} \right\rfloor^{d_0}$ and $n_{2,d_0}^{t,l} = \left\lfloor \frac{n_2^{t,l}}{d_0} \right\rfloor^{d_0}$ then the maximum number of linear regions of functions computed by any ReLU MPNN is lower bounded by*

$$S^{t_0} \frac{\left( \prod_{t=1}^{T} \left( \prod_{l=1}^{L_1^{(t)}} n_{1,d_0}^{t,l} \prod_{l=1}^{L_2^{(t)}} n_{2,d_0}^{t,l} \right) \right)}{n_{1,d_0}^{T,L_1^{(T)}}} \sum_{j=0}^{d_0} \binom{d_T}{j},$$

*where $t_0$ is the number of MPNN layer having max as aggregation operator and for each layer $t$, the index $l$ runs through every layer in $\phi_2^{(t)}, \phi_1^{(t)}$.*

We now sketch some intuition about this result. We can add to $\Phi_1^{(t)}$ (constructed in Algorithm 3 in the Appendix) an initial layer to calculate $\square^{(t)} = \sum$, while the aggregation $\square^{(t)} = \max$ can be seen as a FNN-layer with rank $S$ max activation, thus we can identify $S$ input regions (indicated in red). By setting $\varphi_{\text{Update}}^{(t)}(h_v^{(t-1)}, m_v^{(t)}) = m_v$, we can express the whole MPNN $\chi$ as a FNN applied to input $X$ (details in Algorithm 1, 2 and 3 in the Supplementary). We build on the analysis in [76] to construct intermediate layers that identify $\left\lfloor \frac{n^{t,l}}{d_0} \right\rfloor^{d_0}$ input regions. Our procedure amounts to sequentially folding the input space until the last layer (indicated in blue), and then replicating the hyperplane arrangement in the last layer (indicated in green).

## 4.2 Geometric complexity - Aggregation and Update steps

We now provide a general upper bound for the geometric complexity of ReLU MPNNs when all the weights take integer values - this assumption is mild and holds without loss of generality (details in the Appendix). We call these models *integer-weighted ReLU MPNNs*. A result similar to Proposition 1 establishes the equivalence of integer-weight ReLU MPNNs with tropical rational maps.

For our analysis, we require a technical condition that the network "does not shrink" the representation in the following sense: each intermediate dimension $n_1^{t,l}$ and $n_2^{t,l}$ of $\phi_1^{(t)}$ and $\phi_2^{(t)}$ should be sufficiently

large; and the dimension of the new embedding is at least the dimension of the aggregated message plus the dimension of the previous embedding. These conditions are standard in the analysis using tropical geometry, see e.g., [19].

**Theorem 4.** *[Upper bound on the geometric complexity] Let $\chi : \mathbb{R}^{|V|d_0} \to \mathbb{R}^{|V|d_T}$ be an integer-weight ReLU MPNN. If $\varphi^{(t)}$ satisfies the following conditions for all MPNN layer $t = 1, .., T$*

- $n_2^{t,l} \geq \frac{D}{|V|}d_{t-1}$ *for all* $l = 1, ..., L_2^{(t)}$;
- $n_1^{t,l} \geq d_{t-1}$ *for all* $l = 1, ..., L_1^{(t)}$;
- $n_1^{t,L_1^{(t)}} + d_{t-1} \leq d_t$;

*then the linear degree of $H^{(T)}$ is at most*

$$
\prod_{t=1}^{T} \underbrace{\left( \prod_{l=1}^{L_1^{(t)}-1} \sum_{i=0}^{|V|d_{t-1}} \binom{|V|n_1^{t,l}}{i} \right)}_{\text{from } \Phi_1^{(t)}} \underbrace{\left( \prod_{l=1}^{L_2^{(t)}-1} \sum_{i=0}^{|V|\times d_{t-1}} \binom{Dn_2^{t,l}}{i} \right)}_{\text{from } \Phi_2^{(t)}} \underbrace{\left( \sum_{i=0}^{|V|(\bar{d}_t+d_{t-1})} \binom{|V|d_t}{i} \right)}_{\text{from } \varphi_{Update}^{(t)}} \mathcal{N}_c(\square^{(t)}),
$$

*where* $\mathcal{N}_c(\square^{(t)}) \leq \begin{cases} 1 & \text{if } \square^{(t)} \text{ is sum,} \\ \frac{1}{2}(8S)^{D\tilde{d}_t} & \text{if } \square^{(t)} \text{ is coordinate-wise max/min .} \end{cases}$

We emphasize that, to the best of our knowledge, this the first upper bound for general ReLU message passing architectures. Furthermore, we recover the upper bounds for FNNs and GCNs (with ReLU activations and integer-weights) established in [19] and [21] respectively as special cases.

**Corollary 1.** *The linear degree of an integer-weight FNN is at most $\prod_{t=1}^{T} \left( \sum_{i=0}^{d} \binom{d_t}{i} \right)$.*

**Corollary 2.** *The linear degree of a GCN $\chi$ with $T$ hidden layers, ReLU activation, sum aggregation and integer weight is at most $\prod_{t=1}^{T} \left( \sum_{i=0}^{|V|d} \binom{|V|d_t}{i} \right)$.*

On the other hand, we obtain new bounds for popular GNN models, particularly **GraphSAGE**[13] and **GIN**[15]. For **GraphSAGE**, note that the normalization steps do not change the linear degree.

**Corollary 3.** *Let $\tilde{d}_t$ be the output dimension of $Aggregate_t$ in GraphSAGE [13]. If $d_t \geq d_{t-1} + \tilde{d}_t \geq 2d_{t-1}$ for all MPNN layers $t = 1, ..., T$, then the linear degree of integer-weighted ReLU GraphSAGE described in [13] is upper bounded by*

$$
\mathcal{N}_c(\varphi^T)\ldots\mathcal{N}_c(\varphi^1),
$$

*where*

$$
\mathcal{N}_c(\varphi^t) \leq \begin{cases} \sum_{i=0}^{2|V|d_{t-1}} \binom{|V|d_t}{i} & \text{if the aggregation step is mean,} \\ \left( \sum_{i=0}^{|V|\tilde{d}_{-1}} \binom{|D|\tilde{d}_t}{i} \right) \left( \sum_{i=0}^{|V|(\tilde{d}_t+d_{t-1})} \binom{|V|d_t}{i} \right) & \text{if the aggregation step is pooling.} \end{cases}
$$

**Corollary 4.** *Node embedding of integer-weighted ReLU GIN can be written as [15, Equation 4.1]. Let $n_{t,i}$ be the dimension of each of the intermediate layer in the $MLP^{(t)}$, then its linear degree*

$$
\mathcal{N}_c(\chi) \leq \prod_{t=1}^{T} \left( \prod_{l=1}^{L^{(t)}-1} \sum_{i=0}^{|V|d_t} \binom{|V|n^{t,l}}{i} \right).
$$

### 4.3 Coordinate-wise max vs. sum for aggregation

**Proposition 5.** *[Coordinate-wise max has greater geometric complexity than sum]*

$$
\mathcal{N}(\square^{(t)}) = 1 \qquad\qquad\qquad\qquad\qquad\qquad\qquad \text{if } \square^{(t)} = \sum,
$$
$$
S^{\min\{|V|,D\}\tilde{d}_t} \leq \quad \mathcal{N}(\square^{(t)}) \leq \min\left\{ \sum_{i=0}^{|D|d_{t+1}} \binom{S^2|V|\tilde{d}_t}{i}, S^{|V|\tilde{d}_t} \right\} \quad \text{if } \square^{(t)} = \max .
$$

To establish Proposition 5 we again adapt [76]. Note that if $D \geq |V|$ (the graph is not too sparse, and we have enough messages between the vertices), we have that $\mathcal{N}(\square^{(t)}) = S^{|V|\tilde{d}_t}$ if $\square^{(t)} = \max$. In that case, the geometric complexity will grow polynomially with $S$ (maximum degree).

**Remark 2.** *Interestingly, in [15], the authors point out that if $\phi_1$ and $\phi_2$ are injective, then $\chi$ is as powerful as the WL test. Under that assumption, coordinate-wise is less "expressive" than mean, which is less "expressive" than sum.*

*In contrast, $\phi_1$ and $\phi_2$ are not injective in our case (since ReLU activation is not injective), max is more "expressive" (as measured by the notion of geometric complexity), thus providing another novel insight. Here, the connectivity of the graph plays a particularly important role.*

## 5 New ReLU MPNNs architectures and complexity tradeoffs

Note that while both ReLU FNNs and ReLU MPNNs learn TRSMs/CPLMs (Proposition 1), they might differ vastly in terms of their resource requirements (e.g., the number of layers and parameters). Therefore, we proceed to comparing the complexity of representing a TRSM under the two paradigms (we do not need integer-weight assumption in this section). We simplify our analysis by noting that each component of a TRSM results from the difference of two tropical signomial functions (TSFs) and these TSFs can be computed in parallel using shared layers. Thus, hereafter, we shall focus on TSFs, i.e., functions $f : \mathbb{R}^m \to \mathbb{R}$ of the form $f(x) = \oplus_{i=1}^r c_i \odot x^{\odot \alpha_i}$, where $c_i \in \mathbb{R}, \alpha_i \in \mathbb{R}^m$.

Our idea is to construct as input a clique (i.e., a fully-connected graph) with $m$ nodes $A_1, \ldots, A_m$ and distribute the $r$ monomials (almost) evenly such that each node $A_i$ contains $r' = \lceil \frac{r}{m} \rceil$ monomials (padding with zero monomials if $m \cdot r' > r$). Our construction makes use of a comparison gadget which is already introduced in [19] and [83], and introduces a novel selection gadget (details in the Appendix) which proves to be useful with the permutation equivariance restriction of MPNNs.

The two MPNNs below differ in terms of the way they compare the monomials.

**Global comparison**: Compare $m$ monomials, one from each $A_i$, simultaneously using coordinate-wise max aggregation. Now redistribute (using the selection gadget) the resulting $O(r')$ maximum monomials (each coordinate yields one such monomial) evenly across the nodes, and recur until the maximum across all monomials is obtained.

**Local comparison**: First employ a recursive procedure to compare $r'$ monomials assigned to each $A_i$ locally in order to find a maximum monomial for $A_i$ (breaking ties arbitrarily). Now compare the maximum monomials across the nodes using a single global comparison described above.

One limitation of both these architectures is that they can make no more than $m$ comparisons at a time but $r$ can be much larger. Fortunately, a theorem due to [84], also exploited previously by [83], motivates our construction of a **Constant MPNN** consisting only a constant number of layers to represent any TSF. However, the number of learnable parameters in this MPNN can be exponential in the worst case.

**Remark 3** (Nature of MPNN and FNN). *Without the equivariance constraint on the message component of MPNN, FNN is much more efficient at constructing affine combinations or tropical monomials. On the other hand, thanks to the parallel MP paradigm of MPNN, it is more effective at comparing these monomials, or in other words, increases the geometric complexity of the model. This benefit of MPNN can be seen in Theorem 4, where one has to build much bigger $\Phi_1^{(t)}$ and $\Phi_2^{(t)}$ to reconstruct the result of the parallel $\phi_1^{(t)}$ and $\phi_2^{(t)}$.*

We therefore proceed to our final algorithm that combines the strengths of these two paradigms. **Hybrid MPNN**: We first use a layer of FNN to calculate $r$ tropical monomials and then employ a single layer of MPNN over an $r$-clique with coordinate-max to learn $f$.

**Proposition 6.** *There exist ReLU MPNN algorithms (Local, Global, Constant, and Hybrid that can learn any TSF $f : \mathbb{R}^m \to \mathbb{R}$ with $r$ monomials. Their respective complexity and tradeoffs are summarized in Table 1.*

We provide details about these algorithms and proofs of their correctness as well as complexity in the Supplementary. Local architecture: Algorithm 6 and Proposition 16, Global architecture: Algorithm 7 and Proposition 17 architecture, Constant architecture: Algorithm 8 and Proposition 18, Hybrid architecture: Algorithm 9 and Proposition 19.

| Previously | Message layers | Feedforward layers | Learnable parameters |
|---|---|---|---|
| Deep NN in [19] | None | $\lceil \log_2(r) \rceil + 1$ | $\mathcal{O}(rm)$ |
| Deep NN in [83] | None | $\lceil \log_2(m) \rceil + 1$ | $\mathcal{O}(rm)$ |
| **New (in this work)** | | | |
| Local (Algorithm 6) | 2 | $\lceil \log_2(r/m) \rceil + 5$ | $\mathcal{O}(rm)$ |
| Global (Algorithm 7) | $\lceil \log_2(r) \rceil + 1$ | $3 \lceil \log_m(r) \rceil + 2$ | $\mathcal{O}(rm)$ |
| Constant (Algorithm 8) | 2 | 7 | $\mathcal{O}(mr^{m+2})$ |
| Hybrid (Algorithm 9) | 1 | 1 | $\mathcal{O}(rm)$ |

Table 1: Complexity of representing any tropical signomial function (TSFs) $f : \mathbb{R}^m \to \mathbb{R}$ consisting of $r$ tropical monomials with different architectures. One more layer is required to compute any tropical rational signomial map (TRSM). The four new methods introduced here construct a graph (based on $m$ and $r$) and leverage message passing to efficiently compare these monomials.

## 6 Decision boundary

Lastly, we proceed to characterizing the decision boundary of integer-weighted ReLU MPNNs for classification. We focus on binary classification tasks with a single output to keep the exposition transparent and explicit, albeit one can adapt our construction and analysis accordingly to accommodate multiple classes and outputs. We analyze the decision boundary for both graph and node/edge predictions.

We begin with the analysis for graph classification. Recall from Section 2 that we need an additional readout step $\varphi_{\text{readout}} : \mathbb{R}^{|V| \times d_T} \times \mathbb{R}^{|E| \times d'_T} \to \mathbb{R}^{d_G}$ prior to classification $\eta$, i.e.

$$\chi : \mathbb{R}^{d \times |V|} \times \mathbb{R}^{d' \times |E|} \xrightarrow{\varphi^{(T)} \circ \dots \circ \varphi^{(1)}} \mathbb{R}^{d_T \times |V|} \times \mathbb{R}^{d'_T \times |E|} \xrightarrow{\varphi_{\text{Readout}}} \mathbb{R}^{d_G} \xrightarrow{\eta} \mathbb{R}.$$

Let $\gamma : \mathbb{R} \to \mathbb{R}$ be an injective score function. For $c \in \mathbb{R}$ in $\text{Im}(\gamma)$, we define the decision boundary of $\chi$ as $\mathcal{B} = \{z \in \mathbb{R}^m : \chi(z) = \gamma^{-1}(c)\}$, where $m = d|V| + d'|E|$. The tropical hypersurface $\mathcal{T}(f)$ is precisely the set of points $x$ where $f$ is not linear; i.e., two or more monomials in $f$ achieve the value of $f$ at $x$. We adapt a result from [19, Proposition 6.1] to arrive at the following proposition.

**Proposition 7** (Decision boundary for graph classification). *Let $\chi : \mathbb{R}^{d \times |V|} \times \mathbb{R}^{d' \times |E|} \to \mathbb{R}$ be a ReLU MPNN and $\gamma : \mathbb{R} \to \mathbb{R}$ be an injective score function with $c$ in its range. Then $\chi$ can be viewed as a tropical rational function, $f \oslash g$ and the decision boundary $\mathcal{B}$ defined above divides $\mathbb{R}^m$ into at most $\mathcal{N}(f)$ connected positive regions and at most $\mathcal{N}(g)$ connected negative regions. Furthermore, $\mathcal{B}$ is contained in the tropical hypersurface of a specific tropical polynomial, namely,*

$$\mathcal{B} \subseteq \mathcal{T}(\gamma^{-1}(c) \odot g \oplus f).$$

However, the characterization for node and edge classification requires a more nuanced analysis. We focus on node classification, since the treatment for edges is analogous. In this setting the neural network $\eta : \mathbb{R}^{d_T} \to \mathbb{R}$ is applied to the embedding of each node simultaneously, and a scoring function $\gamma_i : \mathbb{R} \to \mathbb{R}$ is then employed for each node to predict its class based on its score $c_i$. Generally, different scoring functions $\gamma_i$ may be applied, but in practice, we often use the same score function for all the nodes. Viewed individually, the decision boundary for each vertex is similar to Proposition 7; so we pursue a more interesting problem, namely, characterizing the decision boundary of the vertices resulting from a vectorized score function $\Gamma : \mathbb{R}^{|V|} \to \mathbb{R}^{|V|}, [\Gamma(z)]_i = \gamma_i(z)$. However, we immediately hit a roadblock, since choosing a meaningful order for $c_i$ is problematic: the product or subset order is not total whereas the lexical order relies heavily on the ordering of the vertices and thus violates permutation equivariance of the MPNN paradigm. Therefore, we introduce the following notion for the decision boundary.

**Definition 1.** *The decision boundary $\mathcal{B}$ is defined as $\bigcup_{i=1}^{|V|} \mathcal{B}_i$, where $\mathcal{B}_i = \{z \in \mathbb{R}^m : \nu_i(z) = \gamma_i^{-1}(c_i)\}$.*

In order to proceed, we need to generalize the analysis to tropical hypersurfaces. Specifically, we relate $\mathcal{B}$ with the tropical hypersurface of the components of the corresponding tropical rational map. Invoking Proposition 6.1 in [19], we establish the following proposition.

**Proposition 8** (Decision boundary for node classification). *Let $\chi : \mathbb{R}^{d \times |V|} \times \mathbb{R}^{d' \times |E|} \to \mathbb{R}^{|V|}$ be a ReLU MPNN and $\gamma_i : \mathbb{R} \to \mathbb{R}, i = 1, 2, ..., |V|$ be injective score functions with $c_i$ in their range.*

*Then $\chi$ can be viewed as a tropical rational map $F \oslash G$ and its decision boundary is contained in the tropical hypersurface of a specific tropical polynomial, namely,*

$$\mathcal{B} \subseteq \bigcup_{i=1}^{|V|} \mathcal{T}(\gamma^{-1}(c_i) \odot G_i \oplus F_i).$$

## Conclusion, Broader Impact, and Limitations

In this paper, we characterize the class of functions learned by ReLU MPNNs through the lens of Tropical geometry. Thus, beyond previous works that are limited to utilizing tropical geometry in the context of ReLU FNNs, our analysis expands the scope to a widely employed class of GNNs, laying the groundwork for further work on the connections with other machine learning models.

We provide both the lower and upper bounds for the number of linear regions of ReLU MPNNs. The upper bound makes some simplifying assumptions; however, Theorem 4 is still general enough to recover existing bounds for FNNs, GCNs, and provide new bounds for widely used GNN architectures such as GraphSAGE and GIN. Our bound is rather analytical; a numerical approach to counting the number of linear regions can be found in [85]. Adapting the method in [85] for MPNNs, and comparing to our analytical bounds, is an interesting future direction.

We also show that the max aggregation operator is more expressive than the sum operator in terms of geometric complexity of ReLU MPNNs (see Remark 2 on how this result contrasts with the implications of the WL test for injective aggregation functions). We thus showcase the dependence of expressivity on message aggregation operator, and furthermore, the connectivity of the graph structure (max and total degree). It remains open whether spectral quantities such as the spectrum of Laplacian have any effect on the geoemetric complexity of ReLU MPNNs.

The theoretical result on equivalence of class of functions learned by ReLU MPNNs and ReLU FNNs is usually not reflected in practice, where ReLU MPNNs typically outperform FNNs. This motivates our results in Section 5, where we consider several ReLU MPNN architectures to represent CPLMs and compare them to ReLU FNNs. Remark 3, Proposition 6 and Table 1 show that ReLU MPNNs can be more efficient in terms of both the number of learnable parameters as well as the number of layers required in order to be able to represent the same CPLMs. We however, sidestepped other important and practical considerations such as the difference in training time, prediction error and the role of optimisation algorithm (e.g. SGD/Adam). Our focus here has been purely theoretical, and we believe that (at least some of) proposed new architectures would benefit from a comprehensive empirical evaluation. We also characterize the decision boundary for graph classification and node classification, explaining the difference between the two.

Overall, we hope that this work fosters further research on design and analysis of modern deep architectures through the fruitful machinery of tropical geometry.

## Acknowledgments

VG acknowledges support from the Research Council of Finland for the project "Human-steered next-generation machine learning for reviving drug design" (grant decision 342077), the Jane and Aatos Erkko Foundation for the project "Biodesign: Use of artificial intelligence in enzyme design for synthetic biology" (grant 7001703), and a Saab-WASP initiative (grant 411025). TAP wants to thank Johanna Immonen, Jannis Halbey, Negar Soltan Mohammadi, Yunseon (Sunnie) Lee, Bruce Nguyen and Nahal Mirzaie for their discussion, support and wonderful experience during his research internship at Aalto University in 2022.

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

**Technical Appendix - Supplementary Material**

We now describe in detail all the algorithms introduced in the main paper, as well as proofs for all our results. We begin with the results from the section on Tropical algebra.

## Tropical algebra

**Lemma 2.** *Every ReLU FNN $\nu : \mathbb{R}^m \to \mathbb{R}^p$ can be represented by a ReLU MPNN $\chi : \mathbb{R}^{1 \times m} \to \mathbb{R}^{1 \times p}$.*

*Proof.* This is obvious when we set $G$ to be a one-clique (i.e., one vertex graph with a self-edge) and $\phi_2^{(1)} = \nu$, with max or sum aggregation, $\phi_1^{(1)} = \text{id}$, $W_{\text{self}}^{(1)} = 0$ and $W_{\text{neigh}}^{(1)} = \text{Id}_{p \times p}$. $\qquad \square$

**Lemma 3.** *Any ReLU MPNN $\chi : \mathbb{R}^{|V| \times d} \times \mathbb{R}^{|E| \times d'} \to \mathbb{R}^{|V| \times d_{out}} \times \mathbb{R}^{|E| \times d'_{out}}$ can be expressed as a TRSM $F \oslash G : \mathbb{R}^m \to \mathbb{R}^p$, with $m = |V|d + |E|d'$ and $p = |V|d_{out} + |E|d'_{out}$.*

*Proof.* To prove the proposition, we will show that each layer $\varphi^{(t)}$ is a TRSM on its input, thus $\chi$ is a TRSM as well. $\phi_2^{(t)} : \mathbb{R}^{d_{t-1}} \times \mathbb{R}^{d_{t-1}} \times \mathbb{R}^{d'_{t-1}} \times \mathbb{R}^{d'_{t-1}} \to \mathbb{R}^{\tilde{d}_t}$ and $\phi_1^{(t)} : \mathbb{R}^{\tilde{d}_t} \to \mathbb{R}^{\bar{d}_t}$ are ReLU FNNs, hence by [19, Proposition 5.6], they are TRSMs. The aggregation operators (sum and coordinate-wise max for our purpose) are tropical operators pertaining to tropical multiplication and addition respectively. Thus, each component of $\varphi_{\text{Agg}}^{(t)}$ is a TRSF. On the other hand, $\sigma_{\text{Update}}^{(t)}$ is a ReLU activation, the update function $\varphi_{\text{Update}}^{(t)}$, and thus, each MPNN layer $\varphi^{(t)}$ is also a TRSM on its input. $\qquad \square$

***Proof of Proposition 1.*** Note that [19, Corollary 5.3] established the equivalence of $(2), (3), (4)$. Lemma 2 and Lemma 3, imply $(4) \Rightarrow (1)$, and $(1) \Rightarrow (2)$ respectively and hence we are done. $\quad \square$

## Bounds on geometric complexity

We here provide more detailed analysis for the Geometric complexity and the proof of Theorem 3 and Theorem 4. To simplify the analysis, we will assume that we have no edge embedding throughout MPNN layers - the analysis for the most general case follows the same idea. In this analysis, we also assume that inputs of the model follows the same graph $G$. Let $S$ and $D$ be the maximum and total degree of all vertices of $G$.

We will consider the input $H^{(t-1)}$ of each message passing layer $\varphi^{(t)}$ as a stacked vector of edge node embedding $h_v^{(t-1)}$. As we do not want to calculate unused messages between non-adjacency nodes, with the vectorized form (instead of matrix form), we could consider each degree separately and thus reduce the dimension of intermediate layers from $S|V|$ to $D$. Forming the vectorized form requires assumption of an order between the nodes and the induced lexical order for the edges introduced in Section 2. However, we can neutralize this effect by multiplying with the appropriate permutation matrix $P$. We now outline the changes happen in each layer.

1. The vector $H^{(t)} \in \mathbb{R}^{|V|d_t}$ will be the input to $\varphi^{(t)} : \mathbb{R}^{|V|d_{t-1}} \to \mathbb{R}^{|V|d_t}$ ($H^{(0)} = X$).
2. We then apply $\phi_2^{(t)} : \mathbb{R}^{|V|d_{t-1}} \times \mathbb{R}^{|V|d_{t-1}} \to \mathbb{R}^{|V|\tilde{d}_t}$ to every pair of adjacent node embedding $A_i$ and $A_j$ to obtain the message from node $A_i$ to node $A_j$. The idea is to form a big FNN $\Phi_2^{(t)}$ applied on the whole vector $H^{(t-1)}$ to obtain the output $X^{(t)} \in \mathbb{R}^{D\tilde{d}_t}$. The FNN $\Phi_2^{(t)}$ in general depends *depends on $D$ and connectedness of the graph.*
3. After that, we perform the operator $\square^{(t)} = \square_{u \in \mathcal{N}(v)}^{(t)}$ (sum or coordinate-wise max/min) of all the neighboring messages to obtain $Y^{(t)} \in \mathbb{R}^{|V|\tilde{d}_t}$. We will see later that there is a *significant difference* in the two cases of aggregation operator and suggest that the *coordinate-wise max/min* could indeed *increase our geometric complexity.*
4. Analogously, we then form $Z^{(t)} \in \mathbb{R}^{|V|\bar{d}_t}$ by forming a bigger network $\Phi_1^{(t)}$ (based on $\phi_1^{(t)} : \mathbb{R}^{\tilde{d}_t} \to \mathbb{R}^{\bar{d}_t}$), applied to $Y^{(t)}$, i.e. $Z^{(t)} = \Phi_1^{(t)}(Y^{(t)})$.

5. We combine step 2, 3 and 4 to form a big FNN $\Phi_{\text{Agg}}^{(t)}$ applied to $H^{(t-1)}$

6. Lastly, we obtain $H^{(t)} \in \mathbb{R}^{|V| \times d_t}$ by performing the $\varphi_{\text{Update}}^{(t)}$.

We proceed with the first step. The algorithm 1 shows us how to build $\Phi_2^{(t)}$ from $\phi_2^{(t)}$ by replicating the neural network $\phi_2^{(t)}$ for neighboring nodes and putting 0 for the nodes that are not adjacent. Without no edge embeddings, our $\phi_2^{(t)}$ only depends on the node embedding $h_{A_i}^{(t-1)}$ and $h_{A_j}^{(t-1)}$. Our notations from now on always indicate that $\phi_1^{(t)}$, $\phi_2^{(t)}$ are applied to each (pair of) embeddings whereas $\Phi_1^{(t)}$, $\Phi_2^{(t)}$ ($\Phi_3^{(t)}$) are applied to the corresponding stacked vector as a whole.

---

**Algorithm 1** Building $\Phi_2^{(t)}$

---

**Input**: Stacked vector $H^{(t-1)} \in \mathbb{R}^{|V| d_{t-1}}$ of embedding of all vertices.

We will only describe the construction for weight, the bias could be done analogously. Let $W_2^{t,l}$ and $n_2^{t,l}$ be the weight and number of output nodes of the $\ell$-th layer of $\phi_2^{(t)}$ respectively ($n_2^{t,0} = d_{t-1}$). Thus, $W_2^{t,l} \in \mathbb{R}^{n_2^{t,l} \times n_2^{t,l-1}}$. For the first layer $l = 1$, let $W_2^{t,1} = \left[ W_{2,1}^{t,1}, W_{2,2}^{t,1} \right]$, where $W_{2,1}^{t,1}, W_{2,2}^{t,1} \in \mathbb{R}^{n_2^{t,1} \times d_{t-1}}$ correspond to the weight of the first $d_{t-1}$ nodes (of the embedding $h_{A_i}^{(t-1)}$) and last $d_{t-1}$ nodes (of the $h_{A_j}^{(t-1)}$) respectively.

We now define the weight for $\Phi_2^{(t)}$. In the first layer of $\Phi_2^{(t)}$, $\tilde{W}_2^{t,1} \in \mathbb{R}^{D n_2^{t(1)} \times |V| d_{t-1}}$ by breaking it into blocks $[\tilde{W}_2^{t,1}]_{\kappa,\kappa'}$ of size $n_2^{t,1} \times d_{t-1}$. For each edge $\kappa$-th (according to the ordering of edges), assume it is $(A_i, A_j)$, then, we set

$$[\tilde{W}_2^{t,1}]_{\kappa,\kappa'} = \begin{cases} W_{2,1}^{t,1} + W_{2,2}^{t,1} & \text{if } \kappa' = i = j \\ W_{2,1}^{t,1} & \text{if } \kappa' = i \\ W_{2,2}^{t,1} & \text{if } \kappa' = j \\ 0 & \text{otherwise.} \end{cases} \quad (3)$$

The remaining layers $\ell$ are easy to build: let $\tilde{W}_2^{t,l} \in \mathbb{R}^{D n_2^{t,l} \times D n_2^{t,l-1}}$ and $\tilde{W}_2^{t,l} = \begin{bmatrix} W_2^{t,l} & 0 & \dots & 0 \\ 0 & W_2^{t,l} & \dots & 0 \\ 0 & 0 & \dots & W_2^{t,l} \end{bmatrix}$ and applied the activation accordingly.

---

**Proposition 9.** *ReLU FNN $\Phi_2^{(t)}$ in Algorithm 1, applied to $H^{(t-1)}$, will result in the same output as we apply $\phi_2^{(t)}$ separately for neighboring nodes $h_{A_i}^{(t-1)}$, $h_{A_j}^{(t-1)}$ and combine the result.*

*Proof.* We first assume that $\phi_2^{(t)}$ has only one layer and the action of ReLU activation is applied coordinate-wise, we can assume that

$$\phi_2^{(t)}(h_{A_i}^{(t-1)}, h_{A_j}^{(t-1)}) = W_2^{(t),1} \begin{bmatrix} h_{A_i}^{(t-1)} \\ h_{A_j}^{(t-1)} \end{bmatrix} = W_{2,1}^{t,1} h_{A_i}^{(t-1)} + W_{2,2}^{t,1} h_{A_j}^{(t-1)}.$$

There are $D$ edges in total, thus $X^{(t)} = \begin{bmatrix} X_1^{(t)} \\ \dots \\ X_D^{(t)} \end{bmatrix} \in \mathbb{R}^{D n_2^{t,2}}$ and $H^{(t-1)} = \begin{bmatrix} h_{A_1}^{(t-1)} \\ \dots \\ h_{A_{|V|}}^{(t-1)} \end{bmatrix}$. For the $\kappa$-th edge $(A_i, A_j)$, we want $X_\kappa^{(t)} = \phi_2^{(t)}(h_{A_i}^{(t-1)}, h_{A_j}^{(t-1)})$, the formula (3) will yield the desired result.

Then applying the following layers $\Phi_2^{(t)}$ is equivalent to applying the corresponding layers of $\phi_2^{(t)}$ on each result. $\square$

**Proposition 10** (The matrix to take the sum). *If $\square_{u \in \mathcal{N}(v)}^{(t)}$ is sum, then*

$$Y^{(t)} = M X^{(t)},$$

*where $M$ is a matrix that generally depends on the adjacency matrix $A$ of $G$. Thus, it does not change the linear/convex degree and its effect can be absorbed by the first layer of $\Phi_1^{(t)}$.*

*Proof.* We first build $M \in \mathbb{R}^{|V|\tilde{d}_t \times D\tilde{d}_t}$, by breaking it into blocks of size $\tilde{d}_t \times \tilde{d}_t$. Then each block
$$M_{\kappa,\kappa'} = \begin{cases} \mathrm{Id}_{\tilde{d}_t \times \tilde{d}_t} & \text{if edge } \kappa' = (A_\kappa, A_j) \in E \\ 0 & \text{otherwise} \end{cases}.$$

It is easy to see that $Y^{(t)} = MX^{(t)}$, since the matrix $M$ is the same as the adjacency matrix $A$ of $G$ replacing 1 with $\mathrm{Id}_{\tilde{d}_t \times \tilde{d}_t}$. $\qquad\square$

On the other hand, when the operator $\square^{(t)}$ is coordinate-wise max-min, the analysis becomes a little bit more complicated. In particular, we can create a neural network $\Phi_3^{(t)}$ to represent its effect. In the Algorithm 2, we want to make use of the comparing architecture in [83]. A minor blockage is the appearance of the weight $1/2$ in the comparing layer. Fortunately, by scaling the input by a factor of $1/2$ (which does not change the number of linear regions) and scaling the weights by a factor of 2, we could solve this problem. We note that $\Phi_3^{(t)}$ depends heavily on the graph $G$.

---

**Algorithm 2** Building $\Phi_3^{(t)}$

---

**Input**: $X^{(t)} \in \mathbb{R}^{D\tilde{d}_t}$.

We first scale all entries in the input by 1/2. Then we create a FNN that resembles the construction in [83], however, we will not compare two consecutive nodes, but the comparison is done according to the edges of the graph.

---

**Proposition 11.** *If $\square^{(t)}$ is coordinate-wise max/min, then*
$$Y^{(t)} = \Phi_3^{(t)}(X^{(t)}),$$

*where $\Phi_3^{(t)}$ is built according to Algorithm 2.* $\qquad\square$

**Remark 4.** *Here, we surmise that if we have a coordinate-wise max/min aggregation operator, the geometric complexity of our model grows exponentially with $D$ and polynomially with $S$.*

After that, we continue with step 4, where we apply the neural network $\phi_1^{(t)}$ to form the transformation of the aggregated message. Similar to step 2, we would want to build $\Phi_1^{(t)}$ that is applied on the whole $Y^{(t)}$ as a vector. However, this is much easier to form $\phi_1^{(t)}$ than that of $\Phi_2^{(t)}$ as

$$([\Phi_1^{(t)}(Y^{(t)})^\top]_{(i-1)\tilde{d}_t+1,\ldots,i\tilde{d}_t}) = \phi_1^{(t)}(Y^{(t)}_{(i-1)\tilde{d}_t+1,\ldots,i\tilde{d}_t})^\top. \tag{4}$$

---

**Algorithm 3** Building $\Phi_1^{(t)}$

---

**Input:** $Y^{(t)} \in \mathbb{R}^{|V|\tilde{d}_t}$.

Let $W_1^{t,l} \in \mathbb{R}^{n_1^{t,l+1} \times n_1^{t,l}}$ and $\tilde{W}_1^{t,l} \in \mathbb{R}^{|V|n_1^{t,l+1} \times |V|n_1^{t,l}}$ be the weight of the $\ell$-th layer of $\phi_2^{(t)}, \Phi_2^{(t)}$ respectively. Then $\tilde{W}_1^{t,l} = \begin{bmatrix} W_1^{t,l} & 0 & \ldots & 0 \\ 0 & W_1^{t,l} & \ldots & 0 \\ 0 & 0 & \ldots & W_1^{t,l} \end{bmatrix}$ and applied the activation accordingly.

---

Obviously, since each component is applied separately, we have the following proposition.

**Proposition 12.**
$$([\Phi_1^{(t)}(Y^{(t)})^\top]_{(i-1)\tilde{d}_t+1,\ldots,i\tilde{d}_t}) = \phi_1^{(t)}(Y^{(t)}_{(i-1)\tilde{d}_t+1,\ldots,i\tilde{d}_t})^\top. \tag{5}$$

$\qquad\square$

## Lower bound on geometric complexity

***Proof of Theorem 3.*** By Proposition 9, Proposition 13 and Proposition 11, we note that each Aggregation step $\varphi_{Agg}^{(t)}$ can be written as a FNNs $\Phi_{\text{Agg}}^{(t)}$ applied to $H^{(t)}$. Note that $\square^{(t)} = \sum$ can be absorbed by the first layer of $\Phi_1^{(t)}$ (constructed in Algorithm 3). On the other hand, $\square^{(t)} = \max$ can be seen as a max-out layer with rank $S$). Thus, by [76, Theorem 8 proof], it can identify $S$ regions of the input. Then if $\varphi_{\text{Update}}^{(t)}(h_v, m_v) = m_v$ then we can write $H^{(t)} = \Phi_{\text{Agg}}^{(t)}(H^{(t-1)})$. Thus, by [76, Theorem 4], we have the maximum number of linear region of functions computed by any ReLU MPNN is lower bounded by

$$S^{t_0} \frac{\left( \prod_{t=1}^{T} \left( \prod_{l=1}^{L_1^{(t)}} n_{1,d_0}^{t,l} \prod_{l=1}^{L_2^{(t)}} n_{2,d_0}^{t,l} \right) \right)}{n_{1,d_0}^{T,L_1^{(T)}}} \sum_{j=0}^{d_0} \binom{d_T}{j},$$

$\square$

## Upper bound on geometric complexity

**Remark 5.** *If the neural network $\phi_2^{(t)}$ has $n_2^{t,l}$ output nodes in its $\ell$-th hidden layer, then the neural network $\Phi_2^{(t)}$ have $Dn_2^{t,l}$ nodes in its $\ell$-th layer. Therefore, the number of output nodes and hidden layer nodes depend on the connectivity of the network.*

**Proposition 13.** *If the neural network $\phi_1^{(t)}, \phi_2^{(t)}$ has $n_1^{t,l}, n_2^{t,l}$ number of nodes in its $\ell$-th hidden layer then the neural network $\Phi_1^{(t)}, \Phi_2^{(t)}$ have $|V|n_1^{t,l}, |D|n_1^{t,l}$ number of nodes in its $\ell$-th layer. Thus, if*

- $n_2^{t,l} \geq \frac{D}{|V|} d_t$ *for all $l = 1, ..., L_2^{(t)}$;*

- $n_1^{t,l} \geq d_t$ *for all $l = 1, ..., L_1^{(t)}$;*

*then applying [19, Theorem 6.3], we obtain an upper bound for convex degree:*

$$\mathcal{N}_c(\Phi_{\text{Agg}}^{(t)}) \leq \prod_{l=1}^{L_2^{(t)}-1} \sum_{i=0}^{|V|d_t} \binom{Dn_2^{t,l}}{i} \mathcal{N}_c(\square^{(t)}) \prod_{l=1}^{L_1^{(t)}-1} \sum_{i=0}^{|V|d_t} \binom{|V|n_1^{t,l}}{i} \tag{6}$$

*convex linear regions, where*

$$\mathcal{N}(\square^{(t)}) = \begin{cases} 1 & \text{if } \square^{(t)} = \sum \\ \mathcal{O}(S^{D\tilde{d}_t}) & \text{if } \square^{(t)} = \max. \end{cases} \tag{7}$$

*Proof.* If $\square^{(t)}$ is coordinate-wise max/min, then $\Phi_3^{(t)}$ has $\lceil \log_2 S \rceil + 1$ layers. For ease of analysis, we can assume each hidden layer has $D\tilde{d}_t$ nodes (this only increase the convex/linear degree) except for the first layer with at most $2D\tilde{d}_t$ nodes, thus from [19, Theorem 6.3], we have

$$\mathcal{N}_c(\Phi_3^{(t)}) \leq \left( \sum_{i=0}^{D\tilde{d}_t} \binom{2D\tilde{d}_t}{i} \right) \prod_{l=1}^{\lceil \log_2 S \rceil} \sum_{i=0}^{D\tilde{d}_t} \binom{D\tilde{d}_t}{i}$$

$$= 2^{2D\tilde{d}_t - 1} \cdot 2^{D\tilde{d}_t \lceil \log_2 S \rceil},$$

as $\sum_{i=0}^{n} \binom{2n}{i} = 2^{2n-1}$. Using Propositions 9, 11, and 13, we can write the result of $\varphi_{\text{Agg}}^{(t)}$ (i.e. $Z^{(t)}$) as a result of a combined FNN $\Phi_1^{(t)} \circ \Phi_3^{(t)} \circ \Phi_2^{(t)}$ applied to $H^{(t-1)}$. Then applying [19, Theorem 6.3] yields the stated result. $\square$

Thus, we are left with the final step in $\varphi_{\text{Update}}^{(t)}$. If we stack the two vector together, which is now of dimension $|V|\bar{d}_t + |V|d_t$, we can apply [19, Lemma D.4] to prove the following Corollary.

**Corollary 5.** *Let* $\varphi_{Update}^{(t)} = \sigma_{Update}^{(t)} \circ \rho_{Update}^{(t)} : \mathbb{R}^{|V|(\bar{d}_t + d_t)} \to \mathbb{R}^{|V|d_t}$. *If* $|V|(\bar{d}_t + d_t) \leq |V|(d_t)$, *i.e.* $\bar{d}_t + d_t \leq d_t$, *then*

$$\mathcal{N}_c(\sigma_{Update} \circ \rho | (\bar{d}_t + d_t)|V|) \leq \sum_{i=0}^{|V|(\bar{d}_t + d_t)} \binom{|V|d_t}{i}.$$

$\square$

The following results together yield an important Proposition that allows us to analyze each component of $\varphi^{(t)}$ separately before combining them. The following lemma is trivial, but we provide a proof for completeness.

**Lemma 4.** *Let* $F \in Rat(m, p)$ *and* $m' \leq m$, *Then*

$$\mathcal{N}_c(F|m') \leq \mathcal{N}_c(F) \tag{8}$$

*Proof.* We invoke the fact that if $F$ is an affine transformation of $\mathbb{R}^n$ and $A \subset \mathbb{R}^n$ is convex, then the image $F[A]$ is also convex. Thus, a convex linear region in $\mathcal{N}_c(F|d)$ is still a convex and linear region in $\mathcal{N}_c(F)$, and vice versa. On the other hand, some convex linear regions in $\mathbb{R}^n$ do not intersect the affine space, resulting in the inequality. $\square$

We first recall [19, Theorem D.3]: Let $F, G$ in $Rat(m, p)$ and $Rat(m', m)$. Let $H = (h_1, ..., h_p) \in Rat(m', p)$ defined by $h_i := f_i \circ G$ for $i = 1, .., p$. Then

$$\mathcal{N}(H) \leq \mathcal{N}_c(H) \leq \mathcal{N}_c(F|m') \cdot \mathcal{N}_c(G).$$

***Proof of Proposition 2.*** We note that

$$\chi^{(t)} = \varphi^{(t)} \circ \chi^{(t-1)}, \tag{9}$$

thus using [19, Theorem D.3], we have

$$\mathcal{N}_c(\chi^{(t)}) \leq \mathcal{N}_c(\varphi^{(t)} || V|d_{t-1}) \, \mathcal{N}_c(\chi^{(t-1)}). \tag{10}$$

Note that $\varphi^{(t)}(H^{(t-1)}) = \varphi_{Update}^{(t)}(\varphi_{Agg}^{(t)}(H^{(t-1)}), H^{(t-1)})$ thus

$$\mathcal{N}_c(\varphi^{(t)} || V|d_t) \leq \mathcal{N}_c(\varphi^{(t)}) \leq \mathcal{N}_c(\varphi_{Update}^{(t)} | (d_t + d_t)|V|) \, \mathcal{N}_c(\varphi_{Agg}^{(t)}). \tag{11}$$

We can further break the terms $\mathcal{N}_c(\varphi_{Agg}^{(t)})$ down by noticing that $\varphi_{Agg}^{(t)} = \Phi_1^{(t)} \circ \square^{(t)} \circ \Phi_2^{(t)}$, applying in [19, Theorem D.3] multiple times and Lemma 4. $\square$

Thus, bringing all the corollary together and apply [19, Theorem 6.3] recursively, we obtain the following bound for the geometric complexity:

***Proof of Theorem 4.*** The theorem follows from Proposition 2, which tells us that we could analyze the model in steps. Thus, bringing the pieces together, we have

$$\prod_{t=1}^{T} \underbrace{\left( \prod_{l=1}^{L_1^{(t)}-1} \sum_{i=0}^{|V|\bar{d}_t} \binom{|V|n_1^{t,l}}{i} \right)}_{\text{from } \Phi_1^{(t)}} \underbrace{\left( \prod_{l=1}^{L_2^{(t)}-1} \sum_{i=0}^{|V| \times d_{t-1}} \binom{Dn_2^{t,l}}{i} \right)}_{\text{from } \Phi_2^{(t)}} \underbrace{\left( \sum_{i=0}^{|V|(\bar{d}_t+d_{t-1})} \binom{|V|d_t}{i} \right)}_{\text{from } \varphi_{Update}^{(t)}} \mathcal{N}_c(\square^{(t)}),$$

where $\mathcal{N}_c(\square^{(t)}) = \begin{cases} 1 & \text{if } \square^{(t)} \text{ is sum} \\ \frac{1}{2}(8S)^{D\tilde{d}_t} & \text{if } \square^{(t)} \text{ is coordinate-wise max/min} \end{cases}$. $\square$

## 6.1 Consequences of Theorem 4

We first recover the upper bound for FNNs and GCNs (with ReLU activations and integer-weights) established in [19, Theorem 6] and [21, Theorem 4] respectively as special cases. The one for FNN is straightforward. On the other hand, the bound for GCNs requires slightly more work.

***Proof of Corollary 2.*** Firstly, note that GCN can be modelled by setting $\phi_1^{(t)} = Id$, $\square^{(t)} = \text{sum}$, and $\varphi_{\text{Update}}^{(t)} = Z^{(t)}$, where $Z^{(t)}$ is just a stacked vector of $m_{A_i}^{(t)}$. With this simplification, we can in fact have a stronger version of Proposition 2 (going down to input layer):

$$\mathcal{N}_c(\chi) \leq \prod_{t=1}^{T} \mathcal{N}_c(\square^{(t)} \circ \Phi_2^{(t)} || V| d_0)$$

$$\leq \prod_{t=1}^{T} \left( \sum_{i=0}^{|V|d} \binom{|V|d_t}{i} \right) \quad \text{(by Proposition 10, } \square^{(t)} \text{ is just linear transformation).}$$

$\square$

***Proof of Corollary 3.*** In both case, we note that $\phi_1^{(t)} = \text{id}$, thus, we can forget the contribution of $\Phi_1$. If the aggregation operator is mean, we can substitute $\square^{(t)} = \sum$, then $\phi_2^{(t)} = \text{id}$ and $\varphi_{\text{Update}}$ stays the same. Thus, we have

$$\mathcal{N}_c(\varphi^t) \leq \sum_{i=0}^{2|V|d_{t-1}} \binom{|V|d_t}{i} \tag{12}$$

If the aggregation operator is pooling, then $\phi_2^{(t)}$ is a one-layer FNN, $\square^{(t)} = \max$ and $\varphi_{\text{Update}}$ stays the same. Thus, we have

$$\mathcal{N}_c(\varphi^t) \leq \left( \sum_{i=0}^{|V|\tilde{d}_{t-1}} \binom{|D|\tilde{d}_t}{i} \right) \left( \sum_{i=0}^{|V|(\tilde{d}_t + d_{t-1})} \binom{|V|d_t}{i} \right). \tag{13}$$

$\square$

***Proof of Corollary 4.*** According to [15, Equation 4.1], by adding a self-edge to each node in the graph, $t$-th layer of GIN can be written as $\square^{(t)} = \sum$ (or affine transformation - which does not change the geometric complexity), $\phi_2^{(t)} = \text{id}$ and $\phi_1^{(t)} = MLP$ indicated, thus

$$\mathcal{N}_c(\chi) \leq \prod_{t=1}^{T} \left( \prod_{l=1}^{L^{(t)}-1} \sum_{i=0}^{|V|d_t} \binom{|V|n^{t,l}}{i} \right). $$

$\square$

## New ReLU MPNNs architectures and complexity tradeoffs

In most of the following architectures, the aggregated message has the following form,

$$m_{A_i}^{(t)} = \phi_1^{(t)} (\square_{A_j \in \mathcal{N}(A_i)}^{(t)} \phi_2^{(t)} (h_{A_j}^{(t-1)})), \tag{14}$$

i.e., the aggregated message to node $A_i$ depends only on the neighboring nodes $A_j$ (and not $A_i$ itself).

**Lemma 5.** *Let $F \oslash G : \mathbb{R}^m \to \mathbb{R}^p$ be a TRSM. Suppose each component $F_i$ of $F$ can be represented by an FNN/MPNN with $L_{F_i}$ layers, and $G_i$ of $G$ by an FNN/MPNN with $L_{G_i}$ layers. Then $F \oslash G$ can be represented as an $L$-layer FNN/MPNN with $L \leq \max_{i=1}^{p} \{\max\{L_{F_i}, L_{G_i}\}\} + 1$.*

*Proof.* The main idea underlying our proof here is to exploit parallelization. Firstly, for FNNs, we can compute all $F_i$ and $G_i$ in parallel using $\max_{i=1}^{n}\{\max\{L_{F_i}, L_{G_i}\}\}$ layers: we set the block matrix for weights between the different components to zero (for the components that require fewer layers, we can add additional dummy layers with their weights set to the identity matrix $Id$). We then need just one additional layer to compute the difference $F_i \oslash G_i$, again in parallel, for all $i = 1, .., n$. Similarly, for MPNNs, we construct $F_i$ and $G_i$ in parallel following appropriate stacking of the weights of the components. In this case, each additional layer $t$ (for the components with fewer layers) is treated as follows: we set the weights for $\phi_2^{(t)}, \phi_1^{(t)}$ to zero and $W_{\text{self}}^{(t)}$ to identity. Lastly, to compute the difference $F_i$ and $G_i$, we just need one additional MPNN layer where aggregation part is 0 ($\phi_2^{(L)}, \phi_1^{(L)} = 0; \square^{(L)}$ is sum or coordinate-wise max), and $W_{\text{self}}^{(L)} = [Id_{p\times p} \quad -Id_{p\times p}]$. $\qquad\square$

We provide an overview and detailed algorithms for the gadgets here. The *broadcast* gadget sets up an MPNN by replicating its input across nodes of a fully connected graph. It also endows each node with its ROE. The *selection* and *comparison* gadgets are implemented as FFNs: the former is used for partitioning the monomials across the nodes, whereas the latter determines the larger of its two input monomials.

## 6.2 Broadcast gadget

**Broadcast gadget**. Given an input vector $x \in \mathbb{R}^m$, we first construct a fully connected graph with $m$ vertices $A_1, ..., A_m$. The feature vector for $A_1$ is constructed as follows: first $m$ coordinates comprise $x$, the next coordinate is set to 1 (for bias), and the last $m$ coordinates are set to ROE $1_m - e_1$. The feature vectors for all other nodes $A_i$ consist of $m + 1$ zeros followed by $1_m - e_i$. A parameterized message passing layer with weights determined by a target TSF $f$ is then learned to compute all the $r$ monomials $p_1, p_2, \ldots, p_r$ of $f$. We provide all the details, including the operators in Algorithm 4. It can be shown that the embedding of node $A_i$ after broadcasting is $\left[p_1, p_2, ..., p_r, (1_m - e_i)^\top\right]^\top$.

---

**Algorithm 4** Broadcast component: MPNN layer

Build an $m$-clique with vertices $A_1, ..., A_m$ and bidirectional edges, and introduce loops with $m$ self-edges from each vertex $A_i$ to itself.

Prepare node embeddings. For $A_1$, we set its first $m$ coordinates to the input $x$, the next coordinate to 1 for bias, and then the final $m$ coordinates set to its ROE $1_m - e_1$. For other nodes $A_i$, we instead have a vector with first $m$ coordinates set to 0, followed by a single coordinate 0 for bias and finally their respective ROE $1_m - e_i$.

Finally, we construct a layer of message passing. Specifically, $\phi_2^{(1)}$ is a 1-layer FNN with identity activation, no bias, and has the form of Equation 14. Its weight matrix is $\begin{bmatrix} C \in \mathbb{R}^{r\times(m+1)} & 0_{r\times m} \\ 0_{m\times(m+1)} & 0_{m\times m} \end{bmatrix}$, where each row $C_{i,:} = [\alpha_i, c_i] \in \mathbb{R}^{m+1}$. We use the sum aggregation operator and set $\phi_1^{(1)} = Id$. Moreover, $\varphi_{\text{Update}}^{(1)}$ has identity activation, $W_{\text{self}}^{(1)} = \begin{bmatrix} 0_{r\times(m+1)} & 0_{r\times m} \\ 0_{m\times(m+1)} & Id_{m\times m} \end{bmatrix}$, and $W_{\text{neigh}}^{(1)} = Id_{(r+m)\times(r+m)}$.

---

The following result follows immediately from Algorithm 4.

**Proposition 14.** *The embedding of vertex $A_i$ after broadcasting with Algorithm 4 is $\left[p_1, p_2, ..., p_r, (1_m - e_i)^\top\right]^\top$, where $p_i$ is the $i^{th}$ affine combination (or tropical monomial) of $f$.*

*Furthermore, the Broadcast Algorithm 4 uses in total 2 FNN layers across 1 layer of message passing and $(2m + 1)(r + m)$ parameters of which $(m + 1)(r) = \mathcal{O}(rm)$ are learnable.* $\qquad\square$

## 6.3 Selection gadget

We now describe the selection gadget that we invoke to distribute $r$ monomials (almost) evenly among the nodes such that the node $A_i$ gets monomials $p_{\bar{i}} = \{p_k : k \mod m = i\}$. **Selection gadget**. It is a FNN with two layers that acts on the embeddings produced by the Broadcast gadget and utilizes

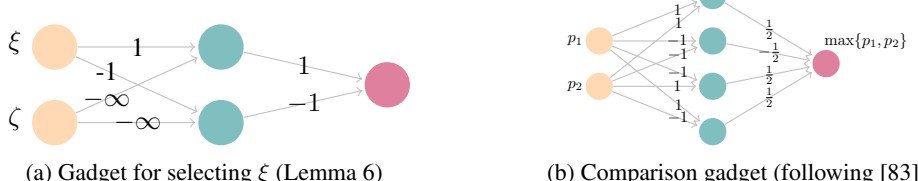

(a) Gadget for selecting $\xi$ (Lemma 6)     (b) Comparison gadget (following [83])

Figure 2: Orange, green, and magenta nodes represent input, hidden (with activation $\max\{\cdot, 0\}$) and output (with activation $\max\{\cdot, -\infty\}$) units respectively. **(Left)** $\zeta$ can be used as a control to either let $\xi$ pass or filter it through the network. Note that in practice a sufficiently small negative weight can be used instead of $-\infty$. **(Right)** A gadget that yields the greater of its two inputs as the output.

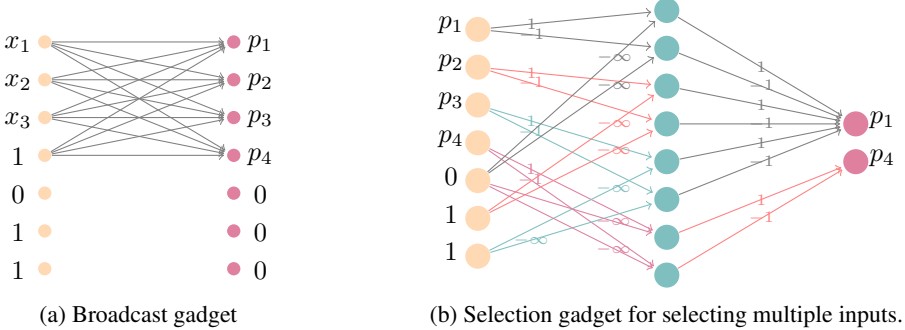

(a) Broadcast gadget     (b) Selection gadget for selecting multiple inputs.

Figure 3: **(Left)** $\phi_2$ in Broadcast gadget (shown here for $m = 3$ and $r = 4$ for node $A_1$) replicates all the monomials $p_1, \ldots, p_r$ across the nodes of a graph $\mathcal{G}$ (not shown). See Algorithm 4 in Appendix for weights and other details. **(Right)** Selection gadget builds on the base gadget from Fig. 2a, and filters out specific monomials at each node of $\mathcal{G}$, yielding a partition of the monomials across nodes.

the following result (Lemma 6) to filter out monomials such that each vertex $A_i$ is left with only the monomials $p_{\bar{i}} = \{p_k : k \mod m = i\}$. The coordinates of each ROE serve as the control variables $\zeta$ for this purpose. Please see Algorithm 5 for details.

**Lemma 6.** *Consider any two-dimensional input with coordinates $\xi \geq 0$ and $\zeta \in \{0, 1\}$. There exists a 2-layer NN with ReLU activations (see Figure 2a) that outputs $\xi$ when $\zeta = 0$, and $0$ when $\zeta = 1$.*

*Proof.* If $\zeta = 0$, then the values at the two nodes in the hidden layer are $\xi$ and $-\xi$ respectively before activation, and $\xi$ and $0$ after activation. Thus, the final output in this case is $\xi$.

On the other hand, if $\zeta = 1$, then the two nodes in the hidden layer take values $-\infty, -\infty$, respectively before activation, and thus $0$ and $0$ after activation. Thus, the final output in this case is $0$. □

**Proposition 15.** *Node $A_i$, after selection with Algorithm 5, only contains the monomials $p_{\bar{i}}$ of $f$.*

*Proof.* Note that this result is just a generalization of Lemma 6, in the sense that we now distribute $r$ monomials into batches of size $m$ (with each monomial playing the role of $\xi$) across the $m$ nodes with coordinates from ROE playing the role of $\zeta$. The two layers return $0$ when $\zeta = 1$ and $\xi$ when $\zeta = 0$. Note that there is only one $0$ in the $i^{th}$ position of ROE $1_m - e_i$ for $A_i$, so $A_i$ ends up with all the monomials $p_{\bar{i}}$ after the second layer. □

We are now ready to describe all the models, namely the Local, Global, Constant and Hybrid algorithms to compute a TSF $f : \mathbb{R}^m \to \mathbb{R}$.

## 6.4 Local MPNN and its complexity

**Proposition 16.** *Local MPNN can learn any TSF $f : \mathbb{R}^m \to \mathbb{R}$ with $r$ monomials. It requires in total $\lceil \log_2(r/m) \rceil + 5$ FNN layers across 2 layers of MPNN, and $\mathcal{O}(rm)$ trainable parameters.*

---

**Algorithm 5** Selection component: FNN layer

---

Input: any vector with ROE. In our case, it is $\left[p_1, p_2, ..., p_r, (1_m - e_i)^\top\right]^\top$

We create a 2-layer neural network as described below.

First layer: no bias, ReLU activation $\max(z, 0)$, and a weight matrix of dimensions $2r \times (r + m)$ with weights

$$
w_{\kappa\kappa'} = \begin{cases} 1 & \text{if } \kappa' \leq r \text{ and } \kappa = 2\kappa' - 1 \\ -1 & \text{if } \kappa' \leq r \text{ and } \kappa = 2\kappa' \\ -\infty & \text{if } \kappa' \geq r \text{ and } \kappa \equiv 2(\kappa' - r), 2(\kappa' - r) - 1 \mod 2m \\ 0 & \text{otherwise.} \end{cases} \tag{15}
$$

Second layer: no bias, identity activation, and a weight matrix of dimensions $r' \times 2r$ with weights

$$
w_{\kappa\kappa'} = \begin{cases} 1 & \text{if } \kappa = \left\lceil \frac{\kappa'}{2m} \right\rceil \text{ and } \kappa' \text{ odd,} \\ -1 & \text{if } \kappa = \left\lceil \frac{\kappa'}{2m} \right\rceil \text{ and } \kappa' \text{ even,} \\ 0 & \text{otherwise.} \end{cases} \tag{16}
$$

---

---

**Algorithm 6** Local MPNN

---

Input: $m$-dimensional input $x$.

First, we perform a broadcast with Algorithm 4.

We need an additional layer of Message Passing. $\phi_2^{(2)}$ will have the form of (14). The first two layers of $\phi_2^{(2)}$ pertain to the selection gadget from Algorithm 5.

We then resort to $\lceil \log_2(r/m) \rceil + 1$ layers of local comparison (simultaneously comparing 2 nodes each time) according to the algorithm in [19] or [83] for $\phi_2^{(2)}$.

$\square^{(2)}$ is coordinate-wise max, $\phi_1^{(2)} = Id$, and $\varphi_{\text{Update}}^{(2)}$ has identity activation with $W_{\text{self}}^{(2)} = 0$ and $W_{\text{neigh}}^{(2)} = Id$.

---

*Proof.* As shown in Proposition 14, each node $A_i$ has the embedding $\left[p_1, p_2, ..., p_r, (1_m - e_i)^\top\right]^\top$ after broadcasting. These embeddings are now filtered by the selection gadget, so by Proposition 15, each node $A_i$ is left with $r' = \lceil \frac{r}{m} \rceil$ monomials $p_{\bar{i}} = \{p_k : k \mod m = i\}$. Thus, using $\lceil \log_2(r') \rceil + 1$ layers of comparison (see, e.g. [83]), we obtain the maximum $\max p_{\bar{i}}$. Max aggregation (coupled with the fact that we have an $m$-clique with self loops) helps us compute $\max_{i=1,...,m} p_{\bar{i}} = \max_{i=1,...,r} p_i = f(x)$. Note that the second MPNN-layer $\varphi^{(2)}$ involves only fixed parameters, and $\phi_2^{(2)}$ comprises $\lceil \log_2(r') \rceil + 3$ FNN layers. $\square$

## 6.5 Global MPNN and its complexity

**Proposition 17.** *Global MPNN can learn any TSF $f : \mathbb{R}^m \to \mathbb{R}$ with $r$ monomials. It requires in total $3 \lceil \log_m(r) \rceil + 2$ FNN-layers across $\lceil \log_m(r) \rceil + 1$ MPNN layers, and $\mathcal{O}(rm)$ trainable parameters.*

*Proof.* As shown in Proposition 14, after the Broadcast component, each node $A_i$ has the embedding $\left[p_1, p_2, ..., p_r, (1_m - e_i)^\top\right]^\top$. Then by Proposition 15, the message from $A_i$ after $\phi_2^{(2)}$ (modified Selection Component) is $p_{\bar{i}} = \{p_k : k \mod m = i\}$, appended by ROE. This vector is of dimension $r' + m = \lceil \frac{r}{m} \rceil + m$. Then, max aggregation (coupled with the fact that we have an $m$-clique with self loops) helps us compute $\max_{i=1,...,m} p_1, ..., p_m, \max_{i=1,...,m} p_{m+1}, ..., p_{2m}$, and so on, and output

---

**Algorithm 7** Global MPNN

---

First, we perform a broadcast with Algorithm 4.

The second layer of MPNN is designed as follows. $\phi_2^{(2)}$ implements the selection gadget from 5 and uses aggregation of the form specified in (14). However, as a placeholder for the ROE, we append $m$ rows with all coordinates set to 0 to the weight matrix for the second layer in Algorithm 5. Thus, the weight matrix is now of dimensions $(r' + m) \times 2r$.

$\square^{(2)}$ is coordinate-wise max and $\phi_1^{(2)} = Id$, $\varphi_{\text{Update}}^{(2)}$ uses identity activation, $W_{\text{neigh}}^{(2)} = Id$, and $W_{\text{self}}^{(2)} = \begin{bmatrix} 0 & 0 \\ 0 & Id_{m \times m} \end{bmatrix}$.

We then repeat this process (replacing $r$ with $r'$ until we are left with only 1 component and the final maximum $f$). This requires $\lceil \log_m(r) \rceil$ additional MPNN layers.

---

the same result for every node $A_i$ besides the ROE which is different for the nodes. Similarly, in the subsequent MPNN layers, invoking the modified selection component and coordinate-wise max yields $\max_{i=1,\ldots,r} p_i = f(x)$. Note that, except for the first MPNN, all the subsequent MPNN layers have only fixed, i.e., non-learnable parameters. Thus in total, we would need $\lceil \log_m(r) \rceil + 1$ MPNN layers, including the first Broadcast layer. $\qquad\square$

### 6.5.1 Constant MPNN and its complexity

---

**Algorithm 8** Constant MPNN

---

First, we perform a broadcast with Algorithm 4.

We only need one additional message passing layer. $\phi_2^{(2)}$ however needs to be modified since for each node $A_i$, we would want to have $l_{z_j,i}$ for each $j = 1,..,q$ suggested by $S_j$ instead of $l_{\bar{i}}$ that Proposition 15 guaranteed.

Specifically, in the first layer of $\phi_2^{(2)}$, we now have $2dr$ hidden nodes ($2d$ nodes for each $l_i$), ReLU activation, no bias, and a weight matrix $\in \mathbb{R}^{2dr \times (r+d)}$ with weights.

$$w_{\kappa\kappa'} = \begin{cases} 1 & \text{if } \kappa' \leq k, \kappa' = \left\lceil \frac{\kappa}{2m} \right\rceil \text{ and } \kappa \text{ odd,} \\ -1 & \text{if } \kappa' \leq k, \kappa' = \left\lceil \frac{\kappa}{2m} \right\rceil \text{ and } \kappa \text{ even,} \\ -\infty & \text{if } \kappa' > k \text{ and } \kappa \equiv 2(\kappa' - k), 2(\kappa' - k) - 1 \mod 2m, \\ 0 & \text{otherwise.} \end{cases} \tag{17}$$

We now choose the linear pieces according to $S_j$. The second layer of $\phi_2^{(2)}$ has $2q$ hidden nodes, identity activation, no bias and weights $\in \mathbb{R}^{2q \times 2dr}$ given by

$$w_{\kappa\kappa'} = \begin{cases} 1 & \text{if } \kappa' = 2m(z_{\mu,i} - 1) + 2i - 1 \text{ for } i = 1,..,m \text{ and } \kappa = 2\mu + 1, \\ 1 & \text{if } 2m(z_{\mu-1,m+1} - 1) + 1 \leq \kappa' \leq 2mz_{\mu-1,m+1}, \kappa' \text{ odd, and } \kappa = 2\mu \\ -1 & \text{if } \kappa' = 2m(z_{\mu,i} - 1) + 2i \text{ for } i = 1,..,m \text{ and } \kappa = 2\mu + 1, \\ -1 & \text{if } 2m(z_{\mu-1,m+1} - 1) + 1 \leq \kappa' \leq 2mz_{\mu-1,m+1}, \kappa' \text{ even, and } \kappa = 2\mu, \\ 0 & \text{otherwise.} \end{cases}$$

For the next layers in $\phi_2^{(2)}$, we take the maximum of node $2\kappa + 1$ and node $2\kappa$ using two layers of the comparison gadget.

The aggregation function is coordinate-wise max, $\phi_1^{(2)} = Id$, $\varphi_{\text{Update}}^{(2)}$ uses identity activation and $W_{\text{self}}^{(2)} = 0$ and $[W_{\text{neigh}}^{(2)}]_j = s_j$.

---

**Proposition 18.** *The Constant algorithm 8 can learn any TSF $f : \mathbb{R}^m \to \mathbb{R}$ with $r$ monomials. It requires in total 7 FNN layers across 2 layers of MPNN, and $\mathcal{O}(mrq)$ learnable parameters.*

*Proof.* z As before, after broadcasting, each node $A_i$ comprises $\left[p_1, p_2, ..., p_r, (1_m - e_i)^\top\right]^\top$. Then, after processing with the first two layers of $\phi_2^{(2)}$, the output for node $A_i$ is $\left[p_{z_{1,i}}, p_{z_{1,m+1}}, ..., p_{z_{q,i}}, p_{z_{q,m+1}}\right]^\top$. Thus, after processed by the last two layers, the message from node $A_i$ is $\left[\max\{p_{z_{1,i}}, p_{z_{1,m+1}}\}, ..., \max\{p_{z_{q,i}}, p_{z_{k,m+1}}\}\right]^\top$. With the max operation, the aggregated message for all nodes is $\left[\max_{i \in S_1} l_i, ..., \max_{i \in S_q} l_i\right]^\top$. Finally, with $s_j$ filling $W_{\text{neigh}}^{(2)}$, the resulting embedding for every node becomes $\sum_{j=1}^{q} s_j(\max_{i \in S_j} l_i)$, which is by [84, Theorem 1], $f(x)$. $\square$

### 6.5.2 Hybrid Architecture and its complexity

---

**Algorithm 9** Hybrid Architecture

**Input:** $m$-dimensional input $x$

We start with a 1-layer FNN with $m$ input nodes and $r$ output nodes for the monomials. It has identity activation and its weight matrix $W \in \mathbb{R}^{r \times m}$ where each row $W_{i,:} = [\alpha_i \in \mathbb{R}^m]$ and its bias is the constants in the tropical monomials, i.e. $b_i = c_i$, thus, yielding the output $y = [p_1 \quad ... \quad p_r]^T$.

After that, we build an $r$-cliques $A_1, ..., A_r$, and bidirectional edges, and introduce loops with $m$ self-edges from each vertex $A_i$ to itself. We then put the initial embeddings $X = y^T = [p_1 \quad ... \quad p_r]$. We then have $\phi_2^{(1)}(h_{A_i}^{(0)}, h_{A_j}^{(0)}) = h_{A_i}^{(0)}$ and Aggregation operation to be coordinate-wise max, $\phi_1^{(2)} = Id$ to form $m_{A_i}^{(1)}$ (which is $f$ now), and put $\varphi_{\text{Update}}^{(1)} = m_{A_i}^{(1)} = f$ .

---

The following Proposition immediately follows.

**Proposition 19.** *The Hybrid algorithm 8 can learn any TSF $f : \mathbb{R}^m \to \mathbb{R}$ with $r$ monomials. It requires in total 1 FNN layer and 1 message passing layer, and $\mathcal{O}(rm)$ learnable parameters.*

