# OpenReview forum: "What do Graph Neural Networks learn? Insights from Tropical Geometry"
_NeurIPS.cc/2024/Conference — NeurIPS 2024 poster_

### Official Review · Reviewer_pwpt · 2024-06-21

**Soundness:** 3
**Presentation:** 1
**Contribution:** 2
**Rating:** 5
**Confidence:** 3

**Summary:**

*Disclaimer*: I did not check proofs carefully and did not read the appendix. I am also by no means an expert in tropical geometry.

This paper studies expressivity of message passing neural networks (MPNNs) through the lens of tropical geometry. The paper has results on equivalences between classes of piecewise linear polynomials, tropical geometric functions, and MPNNs. They also study the expressivity of MPNNs by counting the number of linear regions and expressing types of tropical functions as MPNNs.

**Strengths:**

Proposition 2 and theorems 3 and 4 lay out the main results regarding the expressiveness and the number of linear regions expressible by a MPNN. For theorems 3 and 4, these results have dependence on various quantities that depend on both the graph input and the structure of the MPNN. Nonetheless, a reader can plug these in and get explicit outputs which is nice. Later sections give nice corollaries and extensions of these main results to classification boundaries and expressiveness in the number of parameters.

The paper is largely a theoretical work. Proofs are given in the appendix, but I did not check these. Some of the theory is tied to specific architectures like GraphSAGE and GIN.

**Weaknesses:**

Overall, the math of this paper is something worth publishing, but I think the paper needs to be placed in context and written better. Especially for someone like me who is not in this particular sub-topic of tropical geometry, I found the paper a rather frustrating read.

I wish the authors took a big step back and gave more background into this area and discussed the main results a bit more from a higher level. Many questions exist from reading this that I couldn’t get from just reading the mathematical statements and the main text:
- What does understanding geometric complexity tell us? Does it relate to generalization bounds? When do we want to maximize or minimize this quantity? Or is this paper only concerned with expressiveness?
- How far apart are these bounds from practice?
- Do the expressiveness bounds tell us anything new about GNNs? Are there important classes of functions that it now excludes or includes which are not given by e.g. the WL hierarchy?

From a rhetorical and presentation point of view, I think the authors could benefit greatly from a clearer presentation of mathematical results, a focusing on the main statements, and shortening of key portions of text. I try to outline this in detail in what follows. Much of this is of course my opinion and I am welcome to criticism and feedback.

Proposition 1: This statement seems largely intuitive but is missing details. Can the authors formally define what they mean by equivalence? I.e. the function classes are equal? Do we require the width or number of parameters to be arbitrarily large for this to be true?

Proposition 2:
- This statement is hard to parse as it requires understanding objects like "the maximum convex degree across restrictions of f to different m′-dimensional affine subspaces…” Can the authors explain the importance of this proposition better? All that is stated about its importance is the recursion like property that lets us relate complexity of subsequent layers.
- If this proposition is used to only prove subsequent theorems, I would recommend to remove it here and place in appendix. Again, I have read this subsection two or three times and still cannot parse completely what is going on and why it is necessary to be in the main paper.

Theorem 3:
- The notation of $\mathcal{N}(\chi)$ is a bit confusing here since it does not state the number of linear regions for a function $\chi$, but instead the maximum number of regions for any possible function $\chi$. Perhaps I am missing something though but I feel this notation should be changed if correct to have a maximum in it to be clearer.
- The text before the theorem, especially lines 167 to 169 are hard to parse. This seems to be giving proof intuition though I could not really follow. E.g. what is a max-out layer? What do the authors mean by absorbed in? I don’t think it’s helpful to the reader to refer them to an algorithm in the appendix to understand what appears to be one of the main statements in the paper.
- Can the authors define the notation $n^{t,l}$ before this and preferably even in the theorem statement itself.
- Should the sum in the right hand side in dark green go from $0$ to $d_T$? If correct as is, how does one handle situations where $d_0>d_T$? Also can’t we simplify this part by using the binomial theorem?
- This theorem I think could use some discussion and clarification. It seems that the bound depends rather strongly on the input itself (e.g. number of nodes).  This is in contrast with standard fully connected nets where I don’t see a bound depending on the input since this is fixed.
- Related to the above, how tight is this bound for a random network or one that is found on training?

Theorem 4:
- The integer weight restriction as the authors point out is largely a technical point and I would just state this as a proof technique rather than a whole paragraph to describe it at the start of the section. It is possible I missed the importance of this point here and perhaps it has actual implications or drawbacks that need to be stressed here (I don’t see anything serious though).
- I wish the authors would spend more time explaining the theorems. Similar concerns exist here as in theorem 2. Details are deferred to appendix to explain what is happening in the theorem. Parsing the theorem requires understanding a rather complicated formula. Can we use some big O notation perhaps to clean this up?
- See also notational concerns for theorem 3 that also apply here.


Section 4.4:
- What is PWLM? I assume piecewise linear map? This is not defined though.
- From line 222 and on, I was just lost. Let me try to explain why and perhaps the authors can clear up this section. To begin, the authors say “our idea is to construct a clique (i.e., a fully-connected graph) with m nodes…” What is the purpose of this and why is it even constructed? Then there is a discussion of local and global comparison which I could simply not follow. Words like compare, selection gadget, etc. are used that I do not know what they mean.
- Proposition 6 seems like a simple statement, and I wish the authors would explain more why this is nontrivial or useful to know.

Section 4.5:
- These results seem to be corollaries of theorem 3 and 4 from earlier? Perhaps I am missing something? Is the important part here that the boundary "is contained in the tropical hypersurface of a specific tropical polynomial”?

Conclusion, broader impact, and limitations are considerably short. This paper, especially given it is using tools that may be outside the wheelhouse of the community would really benefit from a richer discussion of limitations and future work.


Literature comments: There are other expressivity hierarchies that the authors could look into which appear more relevant than the WL hierarchy. These include equivariant polynomials (see Puny et al. Equivariant Polynomials for Graph Neural Networks) and homomorphism counts (see Chen et al. Can graph
neural networks count substructures).



Smaller comments:
- Proposition 2 is hard to parse without reading the notation in prior sections and before it. I would prefer more that the most important notation be defined within the formal proposition statement so that it is easier to parse.
- Notation $\mathcal{N}$ is overloaded as defining both linear regions and neighborhood.
- Related to the above, I would request the authors more formally define $\mathcal{N}$ as a number of linear regions in its own definition statement with some discussion about the definition since it seems to be a crucial quantity.
- Line 225: what is “noval”?

**Questions:**

See prior sections.

**Limitations:**

See prior sections.

---

> ### Author Rebuttal · Authors · 2024-08-07
>
> Thank you so much for a detailed review and several excellent suggestions, all of which we will act on. Please see our response below.
>
> **(Geometric complexity (GC) and generalization)**
> GC characterizes the complexity of a neural network to approximate functions.  In particular, a high value indicates high expressivity of the architecture  (since, each linear region could potentially be assigned any of the class labels, independent of the other regions). While expressivity is desirable, it can be at odds with generalization [1], whence we must strive for a good tradeoff. GC thus also has implications for generalization; e.g., see  [2]  for a  bound in the context of gradient flows.
>
> [1] Garg, Jegelka, and Jaakkola. Generalization and representational limits of graph neural networks, 2020.
> [2] Safran, Vardi, and Lee. On the effective number of linear regions in shallow univariate reLU networks: Convergence guarantees and implicit bias, 2022.
>
> `How far apart are these bounds from practice?`
>
> In general, good estimation of GC is non-trivial. That said, we recover the bounds for GCNs [3] as a special case, and [3] had verified empirically that complexity was indeed close to the lower bound.
>
> [3] Chen, Wang, and Xiong. Lower and Upper Bounds for Numbers of Linear Regions of Graph Convolutional Networks, 2022.
>
> **(New results about GNNs vis-a-vis WL)**
> Yes, the WL hierarchy implicitly assumes that all updates are injective, which does not hold with ReLU activations. Thus, WL fails to characterize the exact class of functions that ReLU MPNNs can represent, which we show to be TRSMs. Moreover,  as we note in Remark 2, we provide a novel insight that max is more expressive than sum wrt GC for ReLU MPNNs (in contrast to  a result in [4] where under some injectivity assumptions, sum is more expressive in distinguishing graphs).
>
> [4] Xu, Hu, Leskovec, and Jegelka.  How powerful are graph neural networks, 2019.
>
> **(Proposition 1)**  Yes, equivalence here means that the function classes are equal. We do not require the width or number of parameters to be arbitrarily large: any TRSM can be realised with explicit ReLU MPNN/FNN architectures with fixed number of layers, width, and parameters (Table 1).
>
> **(Proposition 2)** Indeed it allows us to analyze each component of each layer, and combine them to get a bound on the overall GC.  This is important, as explained in Section 4.3., as the choice of aggregation will have an impact on the bound. Moreover, it makes clear the contribution of each component, and captures how modifying any component will affect the bound.
>
> **(Theorem 3)** By $\mathcal{N}(\chi)$ we mean the number of linear region for a particular ReLU MPNN $\chi$, or equivalently a particular continuous piecewise linear map. We do not take the maximum over all possible functions.
>
> A max-out layer is one where the activation function is the max of the inputs [5]. By ``absorbed in" we mean that the network can be extended with another layer of $\Phi$.
>
> We apologise for having to refer to Appendix due to space constraints. Indeed, we wanted to give some intuition for the proofs, however that requires mentioning the big FNNs $\Phi^{(t)}_1, \Phi^{(t)}_2$.  We'll sketch an outline in the main text based on your feedback. We will also define $n^{t, l}$ as suggested.
>
> We believe the formula is correct as is, it goes from $0$ to $d_0$. It builds on the ``hyperplane arrangement'' and ``space folding'' arguments from [5], Zaslavsky (1975), that an arrangement of $n_1$ hyperplanes in $\mathbb{R}^{n_0}$ has at most $\sum_{i=0}^{n_0} \binom{n_1}{i}$.
> We however need to assume that all $n_{1}^{t, l}, n_{2}^{t, l} \geq d_0$
> (in particular,   $d_T \geq d_0$).
>
> Please note the bound for fully connected nets does depend on the dimension of the input. In our case the dimension of the initial embedding is $|V|d_0$, which explains why the number of nodes shows up in our result.
>
> We do not know the tightness of the bound for random networks - it is an interesting open problem.
>
> [5] Montufar, Pascanu, Cho, and Bengio. On the Number of Linear Regions of Deep Neural Networks, 2014.
>
> **(Theorem 4)**  Indeed the assumption of integer weight is a technical condition and can be relegated to Appendix. This would allow us to accommodate other clarifications based on your suggestions.
>
> Apologies - we wanted to be explicit about the role of aggregation and update steps. We could use Stirling's formula to clean up a bit. We will see what we can do about this, and fix any other notational concerns.
>
> **(Section 4.4)** Thanks for catching this: it should be CPLM (continuous piecewise linear map) instead. Sorry for the rather rushed description on gadgets and comparisons - we use these to devise new ReLU MPNN architectures that afford different tradeoffs wrt number of layers and/or parameters.  Proposition 6 precisely elucidates these tradeoffs using Table 1.
>
> **(Section 4.5)** Yes, these are indeed corollaries, and the important part is that the boundary is contained in the tropical hypersurface of a specific tropical polynomial.  Tropical hypersurface is a well-studied object in tropical geometry
> and this allows to study the decision boundary from multiple perspectives.
>
> **(Conclusion)** We hear you, and will do as suggested. In particular, we will point out that some of the technical conditions could be possibly relaxed and a detailed empirical study with novel ReLU MPNN architectures for different kinds of graph inputs (e.g., random graphs) would be helpful.
>
> **(Literature)** Thank you for drawing our attention to these hierarchies. We will be sure to position the mentioned influential works by Puny et al and Chen et al.
>
> We'll take care of the smaller comments about notation and fixing the typo ("noval" should be replaced by novel).
>
> We are grateful for your extremely valuable inputs, and will act on them. Hope all your comments are satisfactorily addressed - we're happy to engage further as well.

---

> > ### Comment · Reviewer_pwpt · 2024-08-08
> >
> > I am at times left more confused with the authors’ answers; so much so that I worry the authors’ results are incomplete and/or wrong. I think the authors were space limited in their responses so let me ask them to expand their response.
> >
> >
> > Let me give a few examples of my confusion and concerns:
> >
> > > [Theorem 3]: By $\mathcal{N}(\chi)$ we mean the number of linear region for a particular ReLU MPNN , or equivalently a particular continuous piecewise linear map. We do not take the maximum over all possible functions.
> >
> > This does not make sense. E.g., set all the weights of the neural network to 0 and there is only one linear region. The theorem cannot possibly hold then.
> >
> >
> > > any TRSM can be realised with explicit ReLU MPNN/FNN architectures with fixed number of layers, width, and parameters (Table 1).
> >
> > ReLU FNN are universal so this cannot make sense that any ReLU MPNN architecture can be realized with fixed parameters just by a counting argument. There has to be some dependence of these on the problem parameters (e.g. input/output dimension).
> >
> > > GC characterizes the complexity of a neural network to approximate functions. In particular, a high value indicates high expressivity of the architecture (since, each linear region could potentially be assigned any of the class labels, independent of the other regions). While expressivity is desirable, it can be at odds with generalization [1], whence we must strive for a good tradeoff.
> >
> > I do not see the connection of generalization to geometric complexity which I assume refers to $\mathcal{N}$. This notion of complexity counts linear regions. Why should I believe this is anything more than a statement about expressivity? This is a worst-case statement as far as I can tell (bounds say at most how many regions there are) and bounds do not seem tight. For example, this quantity can grow exponentially with dimension for example.
> >
> > To argue otherwise for example: is there a nontrivial generalization bound that follows from this notion of complexity?
> >
> > ___
> > I would ask that the authors do the following both in the comments here (and update paper accordingly):
> > - Give formal definitions of $\mathcal{N}(\chi)$ - the in-line description in the text now at line 154 does not make sense to me as mentioned above
> > - Give formal definition of what the authors mean by equivalence of functions
> > - State explicitly relations of geometric complexity and its relations to generalization
> > ___
> > Separate from the examples above, in many places, I also could not follow the authors’ rebuttal and its relation to my comments and questions. I understand this may be due to space limitations, but since the comments are not space limited, I would ask the authors to go back and answer the questions/comments in full.

---

> ### Author Response · Authors · 2024-08-10
> **Detailed comments for clarifications**
>
> Thank you for the discussion, and apologies for any confusion. Indeed, we were hampered by the space constraints, which it seems led to some crucial misunderstandings. Below we provide detailed responses to address these.  We will also update the paper accordingly.
>
> Let's first start with what we mean by the equivalence of functions.
>
> **Equivalence.** Consider the following sets of functions.
>
>  $\mathcal{F}_{\text{ReLU MPNN}}$ : the set of functions represented by all ReLU MPNNs.
>
> $\mathcal{F}_{\text{ReLU FNN}}$ : the set of functions represented by all ReLU FNNs.
>
> $\mathcal{F}_{\text{CPLM}}$ : the set of all continuous piecewise linear maps.
>
> $\mathcal{F}_{\text{TRSM}}$ : the set of all tropical rational signomial maps.
>
> By equivalence, we mean that (apologies for having to break the equations and mathematical descriptions below into separate lines due to math formatting issues here)
>
> $\mathcal{F}_{\text{ReLU MPNN}}$
>
>  $= \mathcal{F}_{\text{ReLU FNN}} $
>
> $=\mathcal{F}_{\text{CPLM}}$
>
>  $=\mathcal{F}_{\text{TRSM}}$
>
> In other words, $f: \mathbb{R}^{m} \to \mathbb{R}^{p}$ is a CPLM
>
> --- iff $f$ is a tropical rational signomial map
>
> --- iff $f$ can be represented by a ReLU FNN $\nu: \mathbb{R}^{m} \to \mathbb{R}^{p}$
>
> --- iff $f$ can be represented by a ReLU MPNN
>
>  $\chi:\mathbb{R}^{|V| \times d} \times \mathbb{R}^{|E| \times d'} \to  \mathbb{R}^{|V| \times d_\text{out}} \times \mathbb{R}^{|E| \times d'_\text{out}}$,
>
> where $m = |V|d +  |E|d'$ and $p=|V|d_\text{out} + |E|d'_\text{out}$
>
> Now, with this equivalence, we can proceed to addressing the following:
>
> `ReLU FNN are universal so this cannot make sense that any ReLU MPNN architecture can be realized with fixed parameters just by a counting argument. There has to be some dependence of these on the problem parameters (e.g. input/output dimension).`
>
> Indeed, as Table 1 in the paper summarises, in order to learn any TRSM with specified input and output dimension, which  consists of $r$ monomials, the complexity (in terms of number of layers, and parameters) of the different ReLU FNN architectures as well as ReLU MPNN architectures depends on the problem parameters (such as input dimension as well as $r$).  By 'fixed' we simply meant that we know the (respective) exact specifications for the proposed ReLU MPNN architectures that realize any TRSM with given input and output dimensions and $r$. In particular, we did not mean that by fixing a particular ReLU MPNN architecture, we can realize all TRSMs independent of input/output dimension or $r$.  Please revisit Proposition 6 and Table 1 for the precise claim apropos of this discussion.
>
>  We next address the concern about $\mathcal{N}(f)$ and  $\mathcal{N}(\chi)$, pertaining to  the following comment
>
> `This does not make sense. E.g., set all the weights of the neural network to 0 and there is only one linear region. The theorem cannot possibly hold then.`
>
> **Clarifying $\mathcal{N}(f)$ and $\mathcal{N}(\chi)$**:
> Apologies, the confusion here does seem to stem from our poor choice of (overloaded) notation, which we will fix as you suggested. For now, to make the argument precise, we first define $\mathcal{N}(f)$.
>
> For a continuous piecewise linear map (CPLM) $f: \mathbb{R}^{m} \to \mathbb{R}^{p}$, we define $\mathcal{N}(f)$ to be the least number of connected regions $C_i$ of $\mathbb{R}^{m}$ such that $F \oslash G|_{C_i}$ is linear. Equivalently, following [1], we can also define $K = \mathcal{N}(f)$ as follows.
>
> $f: \mathbb{R}^{m} \to \mathbb{R}^{p}$ is a continuous piecewise linear map if $f$ is continuous and  there exists a set $\\{f_k: k \in \\{ 1, . . . , K \\}\\}$ of affine functions and nonempty closed subsets $(\Omega_k)_{k=1}^K$ satisfying the following conditions:
>
> $\Omega_i \cap \Omega_j = \emptyset; \quad \bigsqcup_{i=1}^K \Omega_i = \mathbb{R}^m; \quad
> f|_{\Omega_k} = f_k. $
>
> [1]  Alexis Goujon, Arian Etemadi and Michael Unser, On the Number of Regions of Piecewise Linear Neural Networks, 2023.
>
> Now as we claimed above (and proved in the paper), any such CPLM $f$ can be realized by some ReLU MPNN $\chi$ with a particular setting of weights (parameters). However, if we change the weights, the function represented by $\chi$, as well as the number of linear regions, also changes correspondingly.
>
> The particular result gives a lower bound on the maximal number of regions of functions - obtained with different possible settings of weights - that can be represented by a ReLU MPNN. We state this bound below to make things precise:
>
> (Contd...)

---

> ### Author Response · Authors · 2024-08-11
> **Continuation of the clarification thread...**
>
> Consider any ReLU MPNN $\mathbb{R}^{|V |\times d} \to  \mathbb{R}^{|V |\times d_T}$ such that
>
>   (1)  for $t = 1, \dots, T$, $\phi^{(t)}_1$ has $L^{(t)}_1$ layers: intermediate dimension is $n^{(t, l)}_1$ for the $l$-layer ;
>
>   (2) for $t = 1, \dots, T$, $\phi^{(t)}_2$ has $L^{(t)}_2$ layers: intermediate dimension is $n^{(t, l)}_2$ for the $l$-layer;
>
> (3) $t_0$ MPNN-layers have max as the aggregation operator.
>
> Then, the maximal number of linear regions of functions computed by any such ReLU MPNN  is lower bounded by
>
> $S^{t_0}  \left (  \prod_{t=1}^{T-1} \left ( \prod_{l=1}^{L_1^{(t)}}  \left \lfloor \frac{n^{t, l}_1}{d_0} \right \rfloor^{d_0} \prod_{l=1}^{L_2^{(t)}} \left\lfloor \frac{n^{t, l}_2}{d_0} \right\rfloor^{d_0} \right ) \right ) \left (\prod_{l=1}^{L^{(T)}-1} \left\lfloor \frac{n^{T, l}_1}{d_{0}} \right \rfloor^{d_0} \prod_{l=1}^{L^{(T)}-1} \left\lfloor \frac{n^{T, l}_2}{d_{0}} \right \rfloor^{d_0} \right ) \sum_{j=0}^{d_0} \binom{d_T}{j}$,
>
> where $S$ is the maximum degree of input graph $G$.
>
> (Apologies, despite our best efforts, the latex for the above equation does not render here in this environment).
>
> We will update the description of Theorem 3 as above to avoid any possible misinterpretations.
>
> Finally, we sketch below an argument that clarifies the connection of Geometric Complexity (GC) with generalization. We did not include this connection in the original submission, but provide it here based on your comments.
>
> **Geometric complexity and Generalization:**
> We claim that if the geometric complexity is upper bounded by $r$, then the VC-dimension is at most $r$. This immediately translates into a generalization bound (which can be loose though as we explain below) using a classical result from the statistical learning theory for the binary classification setting.
>
> Here's a proof sketch. Since there are (at most) $r$ linear regions, we can select one instance from each linear region to obtain a set of (at most) $r$ instances. We can assign a label for each region independently of others, and moreover, this label can be any of the two possibilities. That is we can "shatter" this set of instances.
>
> Moreover, we cannot shatter $r+1$ (or more) instances. This follows since, otherwise, by pigeonhole principle, at least one region will have two instances.  As a result, we cannot get perfect classification  for this region when these two instances have different labels. Thus, the VC-dimension cannot exceed $r$ (it can, however, be lower than $r$ since the upper bound for the geometric complexity can be loose).
>
> We will add a discussion in the paper to point this out as well.
>
> Thanks for your patience and for your constructive remarks that have helped us elucidate some subtle aspects of this work. We hope that this clarification thread has satisfactorily addressed your concerns. Please let us know if there's something else we can address or elaborate.

---

> > ### Comment · Reviewer_pwpt · 2024-08-11
> >
> > Thank you for your clarifications. I appreciate your attempts to render all these equations in markdown. No problem that they did not all render correctly.
> >
> > I still have concerns about your answers.
> >
> > **On the definition of complexity:** Your definition of $\mathcal{N}(f)$ does not seem to agree with the later statements. You state that the input of $\mathcal{N}$ is a continuous piecewise linear map (CPLM), but later on, your definitions apply to a family of such CPLMs since you take a maximum over all possible weights of the function. So the input to $\mathcal{N}$ should be a family of CPLMs. The way you phrase it here and in the paper is very confusing and simply not correct as far as I can tell.
> >
> >
> > **On VC dimension:** The VC dimension you describe bounded by $r$ is correct but incredibly loose. This bound can grow exponentially with the dimension or number of parameters. In contrast, standard VC bounds for neural networks of bounded width are typically on the order of the number of parameters. E.g. see work by Bartlett et al. Nearly-tight VC-dimension and pseudodimension bounds for piecewise linear neural networks. So I am still very unconvinced this is a more informative notion of generalization.

---

> ### Author Response · Authors · 2024-08-12
>
> Greetings, and thank you for continuing the discussion - we're grateful for your engagement. We're willing to update/qualify notation pertaining to \mathcal{N}, as well as, appropriately position the work by Bartlett et al. in the context of generalization.
>
> **(On the definition of complexity)**:  We acknowledge that overloading $\mathcal{N}$ could be confusing.
>
> One way to take care of this would be as follows. Define $\mathcal{Q}(\mathcal{F}) =  \max_{f \in \mathcal{F}} \mathcal{N}(f)$ for a family of CPLMs $\mathcal{F}$ where each function $f$ corresponds to a different CPLM obtained with a particular choice of the weights. The lower bound in Theorem 3 can then be stated by replacing $\mathcal{N}(\lambda)$ in equation 2 of the paper with $\mathcal{Q}(\mathcal{F_{\lambda}})$ where we define $\mathcal{F}_{\lambda}$ to be the family of functions that can be represented by a specified ReLU MPNN architecture $\lambda$ with all possible settings of the weights.
>
> This would also make everything consistent with the rest of our notation that we adopt from  [Zhang, Naitzat, and Lim. Tropical geometry of deep neural networks, 2018.]
>
> Would you be fine with updating Theorem 3 as above, or do you have any other suggestions to streamline the notation?
>
> **(On VC-dim)**:  Thank you for drawing our attention to the seminal work by Bartlett et al. We believe that our work has taken an important step by showing equivalence between ReLU MPNNs and ReLU FFNs -  both exactly represent CPLMs (though with different tradeoffs in terms of number of parameters etc. as summarised in Table 1) . The generalization analysis by Bartlett et al applies to piecewise linear networks so could potentially be leveraged (applied/adapted/extended) to exploit this connection of ReLU MPNNs with CPLMs.
>
> Indeed, we also agree that bounding VC-dim by $r$ can be very loose. In fact, this was the reason why we sidestepped the question of generalization in the current work, instead choosing to focus on other important aspects where our analysis and results made concrete, non-trivial contributions.
>
> We can add a discussion in the paper along these lines if you think that would be helpful? We also welcome any other suggestions you have in this context.
>
> We hope our response alleviates your concerns. Please do let us know if there's any further clarification we can provide here or update the paper with.  Many thanks!

---

> > ### Comment · Reviewer_pwpt · 2024-08-12
> >
> > The proposed notation is ok with me as long as it is correct. To be clear, this is not about being "willing to update/qualify" the results or "streamline the notation". As the definition is written now, the paper's results are not correct and this must be corrected.
> >
> > I also do not see the point of a discussion about relations to VC dimension when you do not have formal results and the obvious extension of your results produce far worse bounds on FFNs than existing work. I would instead ask you to qualify your statements about generalization since they are to me not convincing and bordering on wrong.

---

> > > ### Author Response · Authors · 2024-08-12
> > >
> > > Thanks - please see our response below.
> > >
> > > `The proposed notation is ok with me as long as it is correct.`
> > > Yes, it is correct. We will update the notation as described in the previous post.
> > >
> > > `I would instead ask you to qualify your statements about generalization since they are to me not convincing and bordering on wrong.`
> > > Could you specify what is wrong with what we said here, or in the paper, about generalization? Only at one place in the main text we mention generalisation in the context of gradient flow [66] (line 91). The referenced paper does use a bound on number of linear regions to get a generalization bound. Do you want us to remove this reference altogether?

---

### Official Review · Reviewer_sZYW · 2024-07-11

**Soundness:** 3
**Presentation:** 3
**Contribution:** 3
**Rating:** 7
**Confidence:** 5

**Summary:**

This paper uses tropical geometry to understand MPNNs in the broader general context of GNNs.  In the face of the WL framework which studies limitations of the GNNs, this paper proposes to use the rich and powerful theory of tropical geometry to uncover their potential.  This paper makes some important contributions in this direction in terms of counts of the number of linear regions analytically and studies particular cases of popular architectures.

**Strengths:**

The paper is quite well-written and well-motivated.  It also lays the ground for further work in this direction where tropical geometry is a powerful tool that has much potential that has so far been limitedly utilized to understand neural networks.

The paper also provides thorough analyses and considerations of the aspects of their theoretical contributions.

**Weaknesses:**

I am very familiar with this area of work and so I was able to understand what the authors meant when claiming to establish equivalence between tropical rational signomial maps, but care needs to be taken to phrase this appropriately because the first equivalence between tropical signomial maps and feedforward neural networks was already established in 2018.

A better literature review is needed on the intersection between tropical geometry and machine learning.  This is a new and promising direction (which I believe that the submission has the potential to make an important contribution) and not much work has been done in this area and so it should be relatively easy to do a comprehensive literature review on this and I am surprised that this was not done.

In general, the references on tropical geometry theory are quite seriously lacking.  It's clear that the proofs of some of the main theoretical contributions rely on known and well-established results from tropical geometry theory and on some of the existing literature at the intersection of tropical geometry and machine learning builds on, I think that it's misleading not to cite these papers and remark explicitly that the approach and idea for the proofs borrow from these existing resources.  It's important that the revision be edited to include this information in order to give appropriate credit where it is due.

**Questions:**

Analytic bounds were given that build on existing theoretical approaches in the literature.  How would the same question to be approached numerically, for example, under experimental training of the network where it will be likely very difficult to derive analytic solutions?

The ideas for future directions of research in the conclusion are not very convincing and seem to be unfounded – can more details please be provided on how "other aspects of tropical geometry" such as "tropical cycles, the Chow group, and Tropical Jacobian" might reveal further insights about the structure of these MPNNs?

Also, this is likely to be a matter of personal taste, but I found the color-coded theoretical results and presentation quite distracting, is this really necessary?

**Limitations:**

The limitations were discussed throughout the paper in the form of remarks and a discussion in the concluding remarks.

---

> ### Author Rebuttal · Authors · 2024-08-07
>
> Many thanks for such a detailed, constructive and thoughtful review. We're grateful for your acknowledgment of the contributions of this work, and share your enthusiasm for leveraging tropical geometry to better understand successful modern architectures.
>
> Below we address all your questions, comments, and suggestions.
>
> `I am very familiar with this area of work... but care needs to be taken to phrase this appropriately because the first equivalence between tropical signomial maps and feedforward neural networks was already established in 2018.`
>
> Thank you for an excellent suggestion.  Indeed, the seminal work of Zhang et al. [1] established the equivalence between ReLU FFNs and tropical relational signomial maps, which we crucially leverage in Proposition 1.  Therefore, we acknowledge this result in the introduction (line 41), at the beginning of Section 3 (line 138), as well as the proof of Proposition 1 (line 513 in the Appendix). Based on your feedback we will state this result prominently at the beginning of Section 3 as a Lemma due to Zhang et al [1], clarifying what the equivalence entails, before we proceed to Proposition 1. We also welcome any other suggestions you might have in this context.
>
> [1] Zhang, Naitzat, and Lim. Tropical geometry of deep neural networks, 2018.
>
> `A better literature review is needed on the intersection between tropical geometry and machine learning. This is a new and promising direction (which I believe that the submission has the potential to make an important contribution) ... it should be ... easy to do a comprehensive literature review. In general, the references on tropical geometry theory are quite seriously lacking.`
>
> Thank you for drawing our attention to this! We will be sure to cite more works at the intersection of tropical geometry and machine learning, and position their contributions. In particular, we will include the following references:
>
> [2] Charisopoulos and Maragos. A Tropical Approach to Neural Networks with Piecewise Linear Activations, 2018.
>
> [3] Maragos, Charisopoulos, and Theodosis. Tropical Geometry and Machine Learning, 2021.
>
> [4]  Alfarra, Bibi, Hammoud, Gaafar, and Ghanem. On the Decision Boundaries of Neural Networks. A Tropical Geometry Perspective, 2023.
>
> [5] Brandenburg, Loho, and Montúfar. The Real Tropical Geometry of Neural Networks, 2024.
>
> [6] Smyrnis and Maragos. Tropical Polynomial Division and Neural Networks, 2019.
>
> [7] Montufar, Ren, and Zhang. Sharp bounds for the number of regions of maxout networks and vertices of minkowski sums, 2022.
>
> [8] Trager, Kohn, and Bruna. Pure and spurious critical points: a geometric study of linear networks, 2019.
>
> [9] Mehta, Chen, Tang, and Hauenstein. The loss surface of deep linear networks viewed through the algebraic geometry lens, 2021.
>
> [10] Grigsby and Lindsey. On transversality of bent hyperplane arrangements and the topological expressiveness of relu neural networks, 2022.
>
> [11] Williams, Trager, Panozzo, Silva, Zorin, and Bruna. Gradient dynamics of shallow univariate relu networks, 2019.
>
> In addition, we welcome (with gratitude) and will include any other relevant references that the reviewer might be able to recommend.
>
> `...proofs of some... contributions rely on known ... results from tropical geometry ... existing literature at the intersection of tropical geometry and machine learning builds on ... it's misleading not to cite these papers and remark explicitly that the approach and idea for the proofs borrow from these existing resources. It's important ...... to give appropriate credit where it is due.`
>
> Thank you for the opportunity to reflect on this. Certainly, as you pointed out, some of our results either invoke directly, or build on, the tools and analysis of influential existing works (e.g., Zhang et al. [1] who laid the foundations for analysis of neural networks via tropical geometry with their pioneering work). This work very much stands on the shoulders of such  giants, and so while we tried our best to acknowledge the contributions of others (e.g., in the proofs in the Appendix), we apologise for any oversights that could potentially be misconstrued in this regard.  We will revisit all the results in this paper, and make sure to acknowledge these contributions in the main text also when we state the propositions.
>
> `Analytic bounds ... build on existing ... literature. How would the same ... be approached numerically, ..., under experimental training of the network where it will be likely very difficult to derive analytic solutions?`
>
> This is an interesting topic for further research. For example, in [12], Serra et al. constructed a method for counting and analyzing, practically, the number of linear regions. Adapting their method for MPNN will require more work, but we believe that it is a promising direction.
>
> [12] Serra, Tjandraatmadja, and Ramalingam. Bounding and Counting Linear Regions of Deep Neural Networks, 2018.
>
> `... future directions ... in the conclusion are not very convincing...  can more details please be provided on how ... "tropical cycles, the Chow group, and Tropical Jacobian" might reveal further insights about the structure of these MPNNs?`
>
> Chow group can be viewed as an analog of homology in algebraic geometry, so we believe it can serve as a type of ”invariance” of a space and its action might  reveal interesting invariants in the setting of ReLU MPNNs.  It also entails questions about tropical cycles and tropical jacobian. However, absent this context (due to space constraints), this might indeed be confusing and unmotivated, so we'd consider removing these directions from the conclusion.
>
> `I found the color-coded theoretical results ... distracting, is this ... necessary?`
> Thank you. We used color-coding to emphasize different components, but will consider removing it.
>
> Thank you very much again, and please let us know if we've sufficiently addressed your concerns. We're also happy to engage further.

---

> > ### Comment · Reviewer_sZYW · 2024-08-11
> > **Acknowledging authors' reply and responding**
> >
> > Dear authors, thank you for your thoughtful and detailed reply to my concerns and comments.  I am happy to read that you have considered them carefully and am satisfied with the proposed changes.  I additionally have read the other reviewers' reports and am satisfied with the authors' responses to them.
> >
> > As mentioned in my review, I would prefer the color-coded theoretical results to be removed.  I also think it is important to remove the directions for future research if more context/details cannot be given (perhaps due to space constraints), because, upon rereading the paper as it stands, these ideas appear to be an ungrounded list of concepts from tropical geometry.
> >
> > I would also add the following references on tropical geometry to the references list:
> > [1] Maclagan, D., & Sturmfels, B. (2021). Introduction to tropical geometry (Vol. 161). American Mathematical Society.
> > [2] Joswig, M. (2021). Essentials of tropical combinatorics (Vol. 219). American Mathematical Society.
> > As long as these changes are implemented, I am happy to keep my positive rating of the paper.

---

> > > ### Author Response · Authors · 2024-08-12
> > >
> > > Greetings, and thank you for your thoughtful and constructive reply. We're glad to hear that your concerns and comments have been addressed.
> > >
> > > We will remove color-coding and the future direction on Chow group as you kindly suggested. Many thanks for also bringing to our attention the works by (Maclagan, D., & Sturmfels, B, 2021) and (Joswig, M., 2021). We will include, and appropriately position, these references in the updated version.
> > >
> > > We're grateful for your support for this work.

---

### Official Review · Reviewer_jpJX · 2024-07-12

**Soundness:** 3
**Presentation:** 3
**Contribution:** 3
**Rating:** 6
**Confidence:** 3

**Summary:**

This paper aims to characterize the class of functions learned by message passing Graph Neural Networks (GNNs) with ReLU activations through the lens of Tropical Geometry. Specifically, it characterizes the functions learned by ReLU-based Message Passing Neural Networks (MPNNs) by establishing their equivalence to ReLU Feedforward Neural Networks (FNNs), Tropical Rational Splines Models (TRSMs), and Continuous Piecewise Linear Maps (CPLMs). The paper provides both lower and upper bounds for the number of linear regions and decision boundaries. Additionally, it compares the expressive power of different aggregation operators, revealing that the coordinate-wise max operator has greater geometric complexity than the sum operator.

**Strengths:**

1. The paper is well-structured and clear.

2. The theoretical foundation is solid and thorough.

3. It provides the first lower and upper bounds for the number of linear regions in ReLU MPNNs.

4. It demonstrates that the max aggregation operator is more expressive than the sum operator in terms of geometric complexity, highlighting how expressivity varies with different message aggregation operators and update functions.

**Weaknesses:**

1. The theoretical analysis may not be directly applicable in practice for graph predictions, as it assumes that the ReLU MPNN processes the same graph structure, which is not typically the case in graph classification tasks.

2. The technical condition used in the analysis—that the dimension of the new embedding is at least the sum of the dimensions of the aggregated message and the previous embedding—may be restrictive and not hold for general MPNN architectures.

3. It is unclear how the input graph's spectral properties affect the geometric complexity of ReLU MPNNs within the established bounds.

4. While the theoretical results indicate that ReLU MPNNs are less expressive or as expressive as ReLU FNNs, in practice, ReLU MPNNs significantly outperform FNNs in graph learning tasks. This discrepancy suggests the theoretical results may not fully explain the practical successes of ReLU MPNNs.

Other Comments:

1. Line 123: $x_d^{a_m}$ should be $x_m^{a_m}$.

2. What does $D$ stand for in Theorem 4?

3. It may be better to use different notations to represent the set of neighboring vertices ($\mathcal{N}(v)$) and the linear degree ($\mathcal{N}(f)$) for a piecewise linear function to avoid ambiguity.

**Questions:**

Please refer to the Weaknesses.

---

> ### Author Rebuttal · Authors · 2024-08-07
>
> Thank you for your thoughtful and constructive feedback. We're glad to note your recognition of several contributions and strengths of this paper.  We address your comments, concerns, and suggestions below.
>
> `The theoretical analysis ... assumes that the ReLU MPNN processes the same graph structure, ....`
>
> We agree with the reviewer. This is a good observation, and indeed we assume the same graph structure for our analysis while deriving the upper bound on geometric complexity. We suspect that this is a technical artefact that somewhat simplifies our analysis (previous work on number of linear regions of GCNs [1] also made a similar assumption), but retains the essential aspects of the typical (i.e., the more general different graph structures) setting with respect to the role played by the aggregation and update steps. We indeed hope that this assumption can be relaxed, or removed altogether, in some future work.
>
> [1] Chen et al. Lower and upper bounds for numbers of linear regions of graph convolutional networks, 2022.
>
> `The technical condition ... that the dimension of the new embedding is at least the sum of the dimensions of the aggregated message and the previous embedding—may be restrictive ...`
>
> We agree with the reviewer on this as well.  This restriction generally arises in the analysis of geometric complexity (in particular, it showed up in the analysis of tropical geometry of ReLU FFNs [2] as well)  to ensure an amenable arrangement of hyperplanes, ruling out the pathological possibilities.
>
> [2] Zhang et al. Tropical geometry of deep neural networks, 2018.
>
> `It is unclear how the input graph's spectral properties affect the geometric complexity of ReLU MPNNs within the established bounds.`
>
> Indeed, while our bounds reveal the dependence of some input graph properties such as max degree, sum etc., they do not involve spectral quantities such as the spectrum of Laplacian. It's an interesting future direction to investigate whether such quantities have any effect on the geometric complexity of ReLU MPNNs.
>
> `While the theoretical results ... ReLU MPNNs are less expressive or as expressive as ReLU FNNs, in practice, ReLU MPNNs significantly outperform FNNs in graph learning tasks. This discrepancy... may not fully explain the practical successes of ReLU MPNNs.`
>
> Thank you for the opportunity to clarify this question that motivated most of the second half of our work, where we consider several ReLU MPNN architectures to represent continuous piecewise linear maps and compare them to ReLU MPNNs. While the first half of our work showed an equivalence between ReLU MPNNS and ReLU FFNs in terms of the class of functions that they can represent,  our subsequent analysis  (especially, Remark 3, Proposition 6,  and Table 1 in the paper) suggests why ReLU MPNNs might be more effective in practice. In particular, as Table 1 summarises, ReLU MPNNs outperform ReLU FNNs in terms of both the number of learnable parameters as well as the number of layers required in order to be able to represent the same continuous piecewise linear map. That said, we sidestepped the role of optimisation algorithm (e.g., SGD/Adam), which is another consideration that might also add to this  discrepancy between ReLU MPNNs and ReLU FNNs in practice.  Based on your feedback, we will make this clear.
>
> `Other Comments:
> Line 123: should be .
> What does  stand for in Theorem 4?
> It may be better to use different notations to represent the set of neighboring vertices () and the linear degree () for a piecewise linear function to avoid ambiguity.`
>
> Many thanks - we'll incorporate all your suggestions. Indeed, in line 123, $x^{a_m}_d$ should be $x^{a_m}_m$. In Theorem 4, $D$ stands for the sum of degrees of all the vertices. We would also modify the notation to avoid confusion.
>
> We're grateful for your review, and hope our response has satisfactorily addressed your questions and concerns. Thank you very much!

---

> > ### Comment · Reviewer_jpJX · 2024-08-12
> >
> > Thank you for your response. I think my score still accurately reflects my belief in the paper and I retain the score.

---

### Author Response · Authors · 2024-08-14
**A word of appreciation**

We would like to thank  the reviewers for their constructive feedback, and chairs (area, senior area, program) for their service.

As detailed below in our individual responses to the reviewers, we've acted on all of the reviewers' comments and suggestions, and tried our best to address their concerns and questions, even the minor ones.  We commit to reflecting the promised edits (e.g., adding related references, fixing a notation in one of the results, etc.) in the updated version.

We believe acting on the reviewers' feedback has helped us consolidate the contributions, and reinforced the strengths of this work.

Many thanks!

---

### Decision · Program_Chairs · 2024-09-25

**Decision:**

Accept (poster)

**Comment:**

The paper presents a new and fundamental approach to understanding the expressivity of message-passing Graph Neural Networks (GNNs) using the lence of tropical geometry. The paper establishes an equivalence between ReLU GNNs and various classes of continuous piecewise linear maps, contributing new bounds on the geometric complexity of popular GNN architectures. The paper notably provides general upper and lower bounds on the number of linear regions and decision boundaries in such networks.

While the paper makes solid theoretical contributions, the presentation of the results could benefit from substantial improvement. The material is dense and may be challenging for readers (especially for readers without a strong background in tropical geometry). Moreover, better attribution of previous work in tropical geometry is expected. Additionally, the practical implications of the theoretical results remain unclear, with no experimental validation provided to assess how these theoretical bounds translate to real-world applications.

Despite these shortcomings, the reviewers unanimously recognize the novelty and importance of the results. The authors are encouraged to revise the manuscript to improve clarity. Future work investigating the practical relevance of the results would be very welcome.